# Noether's Razor: Learning Conserved Quantities

**Tycho F. A. van der Ouderaa**
Imperial College London
London, UK

**Mark van der Wilk**
University of Oxford
Oxford, UK

**Pim de Haan**
CuspAI
Amsterdam, NL

## Abstract

Symmetries have proven useful in machine learning models, improving generalisation and overall performance. At the same time, recent advancements in learning dynamical systems rely on modelling the underlying Hamiltonian to guarantee the conservation of energy. These approaches can be connected via a seminal result in mathematical physics: Noether's theorem, which states that symmetries in a dynamical system correspond to conserved quantities. This work uses Noether's theorem to parameterise symmetries as learnable conserved quantities. We then allow conserved quantities and associated symmetries to be learned directly from train data through approximate Bayesian model selection, jointly with the regular training procedure. As training objective, we derive a variational lower bound to the marginal likelihood. The objective automatically embodies an Occam's Razor effect that avoids collapse of conservation laws to the trivial constant, without the need to manually add and tune additional regularisers. We demonstrate a proof-of-principle on $n$-harmonic oscillators and $n$-body systems. We find that our method correctly identifies the correct conserved quantities and U$(n)$ and SE$(n)$ symmetry groups, improving overall performance and predictive accuracy on test data.

## 1 Introduction

Symmetries provide strong inductive biases, effectively reducing the volume of the hypothesis space. A celebrated example of this is the convolutional layer embedding translation equivariance in neural networks, which can be generalised to other symmetry groups [Cohen and Welling, 2016].

Meanwhile, physics-informed machine learning models [Greydanus et al., 2019, Cranmer et al., 2020], typically relying on neural differential equations [Chen et al., 2018], embed constraints known from classical mechanics into model architectures to improve accuracy on physical dynamical systems.

Rather than strictly constraining a model to certain symmetries, recent works have explored whether invariance and equivariance symmetries in machine learning models can also be automatically learned from data. This often relies on separate validation data [Maile et al., 2022], explicit regularisers [Finzi et al., 2021] or additional outer loops [Cubuk et al., 2018]. Alternatively, we can take a Bayesian approach where we embed symmetries into the prior and empirically learn them through Bayesian model selection [van der Wilk et al., 2018, Immer et al., 2022, van der Ouderaa et al., 2022].

We propose to use *Noether's theorem* [Noether, 1918] to parameterise symmetries in Hamiltonian machine learning models in terms of their conserved quantities. To do so, we propose to symmetrise a learnable Hamiltonian using a set of learnable quadratic conserved quantitites. By choosing the conserved quantities to be quadratic, we can find closed-form transformations that can be used to obtain an unbiased estimate of the symmetrised Hamiltonian.

Secondly, we phase symmetries implied by conserved quantities in the prior over Hamiltonians an leverage the *Occam's razor* effect of Bayesian model selection [Rasmussen and Ghahramani, 2000, van der Wilk et al., 2018] to learn conserved quantities and their implied symmetries directly from train data. We derive a practical lower bound using variational inference [Hoffman et al., 2013]

38th Conference on Neural Information Processing Systems (NeurIPS 2024).

resulting in a single end-to-end training procedure capable of learning the Hamiltonian of a system jointly with its conserved quantities. As far as we know, this is the first case in which Bayesian model selection with variational inference is successfully scaled to deep neural networks, an achievement in its own right, whereas most works so far have relied on Laplace approximations [Immer et al., 2022].

Experimentally, we evaluate our *Noether's razor* method on various dynamical systems, including $n$-simple harmonic oscillators and $n$-body systems. Our results suggest that our method is indeed capable of learning the conserved quantities that give rise to correct symmetry groups for the problem at hand. Quantitatively, we find that our method that learns symmetries from data matches the performance of models with the correct symmetries built-in as oracle. We outperform vanilla training, resulting in improved test generalisation and predictions that remain accurate over longer time periods.

## 2 Background

### 2.1 Hamiltonian mechanics

Hamiltonian mechanics is a framework that describes dynamical systems in *phase space*, denoted $\mathcal{M} = \mathbb{R}^M$, with $M$ even. Phase space elements $(\boldsymbol{q}, \boldsymbol{p}) \in \mathcal{M}$ follow *Hamiltonian equations of motion*:

$$\dot{q}_i = \frac{\partial H}{\partial p_i}, \quad \dot{p}_i = -\frac{\partial H}{\partial q_i} \tag{1}$$

where the Hamiltonian $H : \mathcal{M} \to \mathbb{R}$ is an *observable*[1], which are smooth functions on the phase space, that corresponds to the energy of the system. It is often simpler to write $\boldsymbol{x} = (\boldsymbol{q}, \boldsymbol{p})$, so that we have: $\dot{\boldsymbol{x}} = J\nabla H$ and $J = \begin{bmatrix} 0 & I \\ -I & 0 \end{bmatrix}$, where $I$ is the identity matrix, $J$ is called the *symplectic form* and $\nabla H = \nabla_{\boldsymbol{x}} H(\boldsymbol{x})$ is the gradient of phase space coordinates.

*Example: $n$-body problem in 3d.* If we consider a $d=3$ dimensional Euclidean space containing $n$ bodies, our position and velocity spaces are each $\mathbb{R}^{3n}$ making up phase space $\mathcal{M} = \mathbb{R}^{2 \cdot 3n}$. Our Hamiltonian $H : \mathbb{R}^{3n} \times \mathbb{R}^{3n} \to \mathbb{R}$, which in this case is a separable function $H(q, p) = K(q) + P(p)$ of kinetic energy $K(q) = \sum_i m_i ||p||^2/2$ and the potential energy $P(p) = \sum_{i \neq j} Gm_i m_j/||q_i - q_j||$ where $m_i$ is the mass of a body $i$ and $G$ is the gravitational constant.

### 2.2 Learning Hamiltonian mechanics from data

We can model the Hamiltonian from data [Greydanus et al., 2019, Ross and Heinonen, 2023, Tanaka et al., 2022, Zhong et al., 2019]. Concretely, we are interested in a posterior over functions that the Hamiltonian can take $p(H_{\boldsymbol{\theta}} \mid \mathcal{D})$, conditioned on trajectory data $\mathcal{D} = \{(\boldsymbol{x}_t^n, \boldsymbol{x}_{t'}^n)\}_{n=1}^N$ sampled from phase space at different time points $(t, t')$, or time difference $\Delta t = t' - t$. Given a new data point $\boldsymbol{x}_t^*$, we would like to make predictions $p(\boldsymbol{x}_{t'}^* | \boldsymbol{x}_t^*, H_{\boldsymbol{\theta}}, \mathcal{D})$ over phase space trajectories into the future $t'$.

**Hamiltonian neural networks** Hamiltonian neural networks [Greydanus et al., 2019, Toth et al., 2019, Rezende et al., 2019] model the Hamiltonian $H$ using a learnable Hamiltonian $H_{\boldsymbol{\theta}} : \mathcal{M} \to \mathbb{R}$ parameterised by $\boldsymbol{\theta} \in \mathbb{R}^P$. With a straightforward Gaussian likelihood $p(\boldsymbol{x}_{t'} | \boldsymbol{x}_t, \boldsymbol{\theta}) = \mathcal{N}(\boldsymbol{x}_{t'} | \boldsymbol{x}_t + J\nabla H_{\boldsymbol{\theta}}(\boldsymbol{x}_t)\Delta t, \sigma_{\text{data}}^2 \boldsymbol{I})$ with a small observation noise $\sigma_{\text{data}}^2$, a maximum likelihood fit can be found by minimising the negative log-likelihood $\boldsymbol{\theta}_* = \arg\min_{\boldsymbol{\theta}} \sum_i \sum_t -\log p(\boldsymbol{x}_{t+\Delta t}^i | \boldsymbol{x}_t^i, \boldsymbol{\theta})$ on minibatches of data using stochastic gradient descent. The mean of this likelihood represents a single Euler integration step (Sec. 2.1 of David and Méhats [2023]), which bounds the possible accuracy of the fit to the true Hamiltonian $H$. In practice, we may replace this by more accurate differentiable numerical integrators [Kidger, 2022].

### 2.3 Noether's theorem

The theorem of [Noether, 1918], here presented in the Hamiltonian formalism [Baez, 2020, Arnold, 1989], links the concepts of an observable being conserved, to the Hamiltonian being invariant to the symmetries generated by an observable.

---

[1]The term *observable* in classical mechanics should not be confused with the statistical notion of a variable being observed or not. In fact, we will model observables as latent variables that are not observed.

**Conserved quantity**    Let $\mathcal{O}$ be the set of observables, which are smooth real-valued functions $\mathcal{M} \to \mathbb{R}$ on the phase space. Given a trajectory $\boldsymbol{x}(t)$ generated by the Hamiltonian $H$, we can compute the variation of an observable $O \in \mathcal{O}$ in time via the chain rule and Hamilton's equations of motion (Equation (1))

$$\frac{\mathrm{d}O}{\mathrm{d}t} = \sum_i \frac{\partial O}{\partial q_i}\dot{q}_i + \frac{\partial O}{\partial p_i}\dot{p}_i = \sum_i \frac{\partial O}{\partial q_i}\frac{\partial H}{\partial p_i} - \frac{\partial O}{\partial p_i}\frac{\partial H}{\partial q_i} = \{O, H\}, \tag{2}$$

where the last equality defines the *Poisson bracket* $\{\cdot, \cdot\} : \mathcal{O} \times \mathcal{O} \to \mathcal{O}$. The Poisson bracket relates to the symplectic form via $\{O, H\}(\boldsymbol{x}) = \nabla O(\boldsymbol{x}) \cdot J \nabla H(\boldsymbol{x})$. An observable that does not change along any trajectory is called a *conserved quantity*. As we can see from Equation (2), an observable $O$ is conserved if and only if $\{O, H\} = 0$.

From two conserved quantities $O, O' \in \mathcal{O}$, we can create a new conserved quantity by linear combination $\alpha O + \beta O' \in \mathcal{O}$ with coefficients for $\alpha, \beta \in \mathbb{R}$, which is conserved because the Poisson bracket is linear in both arguments. Also, we can take the product $OO' \in \mathcal{O}$, with $(OO')(\boldsymbol{x}) = O(\boldsymbol{x})O'(\boldsymbol{x})$, which is conserved because the Poisson bracket satisfies Leibniz's law of differentiation $\{OO', H\} = \{O, H\}O' + O\{O', H\}$. Finally, the Poisson bracket of the conserved quantities $\{O, O'\} \in \mathcal{O}$ is also conserved, because of the Jacobi identity.

**Symmetries generated by observables**    Referring back as to the Hamiltonian equations of motion in Equation (1), note that these equations work not just for the Hamiltonian $H \in \mathcal{O}$ of the system, but for *any* observable $O \in \mathcal{O}$. So given any starting point $x_0$, we can generate a trajectory $\boldsymbol{x}(\tau)$ satisfying

$$\boldsymbol{x}(0) = \boldsymbol{x}_0 \qquad\qquad \dot{\boldsymbol{x}}(\tau) = J \nabla O(\boldsymbol{x}(\tau)). \tag{3}$$

We have used a different symbol to not conflate the ODE time $\tau$ with regular time $t$ of the trajectory generated by the Hamiltonian. Denote the flow associated to this ODE generated by observable $O$ by $\Phi_O^\tau : \mathcal{M} \to \mathcal{M}$, mapping $\boldsymbol{x}_0$ to $\Phi_O^\tau(\boldsymbol{x}_0) = \boldsymbol{x}(\tau)$. Note that any ODE flow satisfies $\Phi_O^0 = \mathrm{id}_\mathcal{M}$ and $\Phi_O^{\tau+\kappa} = \Phi_O^\tau \circ \Phi_O^\kappa$. Hence, the observable $O$ generates a one-dimensional group $\mathcal{G}_O$, parametrized by $\tau$, that is a subgroup of the group $\mathrm{Diff}(\mathcal{M})$ of diffeomorphisms $\mathcal{M} \to \mathcal{M}$.

**Theorem 1** (Noether). *The observable $O \in \mathcal{O}$ is a conserved quantity on the trajectories generated by Hamiltonian $H \in \mathcal{O}$ if and only if $H$ is invariant to $\mathcal{G}_O$, meaning that for all $\tau \in \mathbb{R}$, $H \circ \Phi_O^\tau = H$.*

*Proof.* By reasoning analogous to that in Equation (2), the value of the Hamiltonian changes under the flow generated by observable $O$ as $\frac{\mathrm{d}H}{\mathrm{d}\tau} = \{H, O\}$. Noting that the Poisson bracket is anti-symmetric, we have that: $O$ is a conserved quantity $\iff \{O, H\} = 0 \iff \{H, O\} = 0 \iff H$ is invariant to the flow generated by $O$. $\qquad\square$

## 2.4  Automatic symmetry discovery

Symmetries play an important role in machine learning models, most notably group invariance and equivariance constraints [Cohen and Welling, 2016]. Instead of having to define symmetries explicitly in advance, recent attempts have been made to learn symmetries automatically from data. Even if learnable symmetries can be differentiably parameterised, learning them can remain difficult as symmetries act as constraints on the functions a model can represent and are, therefore, not encouraged by objectives that solely optimise train data fit. As a result, even if a symmetry would lead to better test generalisation, the training collapses into selecting no symmetry. Common ways to overcome this are designing explicit regularisers that encourage symmetry [Benton et al., 2020, van der Ouderaa et al., 2022], which often require tuning, or use of validation data [Alet et al., 2021, Maile et al., 2022, Zhou et al., 2020]. Learning symmetries for integrable systems was proposed in [Bondesan and Lamacraft, 2019], whereas our framework works more generally also for non-integrable systems, such as the 3-body problem. Recent works have demonstrated effectivity of Bayesian model selection to learn symmetries directly from training data. This works by optimising the marginal likelihood, which embodies an Occam's razor effect that trades off data fit and model complexity. For Gaussian processes, the quantity can often be computed in closed-form [van der Wilk et al., 2018], and can be scaled to neural networks through variational inference [van der Ouderaa and van der Wilk, 2021] and linearised Laplace approximations [Immer et al., 2022].

# 3 Symmetrising Hamiltonians with Conserved Quantities

Our method introduced in the next section will learn the Hamiltonian of a system together with a set of conserved quantities. First, in this section we discuss how the learned conserved quantities will be parametrised, and how we can make the Hamiltonian invariant to the symmetry generated by conserved quantities.

## 3.1 Parameterising conserved quantities

In this work, we limit ourselves to modelling up to a fixed maximum number of $K$ conserved quantities $C_{\boldsymbol{\eta}}^1, C_{\boldsymbol{\eta}}^2, \ldots, C_{\boldsymbol{\eta}}^K : \mathcal{M} \to \mathbb{R}$ are observables parameterised by *symmetrisation parameters* $\boldsymbol{\eta}$, to distinguish them from the *model parameters* $\boldsymbol{\theta}$ parameterising the Hamiltonian scalar field.

In this paper, we consider quadratic conserved quantities of the form $C_{\boldsymbol{\eta}}(\boldsymbol{x}) = \boldsymbol{x}^T \boldsymbol{A} \boldsymbol{x}/2 + \boldsymbol{b}^T \boldsymbol{x} + c$. As we use the conserved quantities only through their gradients, the constant is arbitrary and can be ignored. The learnable symmetrisation parameters are thus $\boldsymbol{\eta} = \{\boldsymbol{A}, \boldsymbol{b}\}$, for a symmetric matrix $\boldsymbol{A}$. A quadratic conserved quantity $C$ generates a symmetry transformation whose scalar field $\dot{\boldsymbol{x}} = J\nabla C(\boldsymbol{x}) = J\boldsymbol{A}\boldsymbol{x} + J\boldsymbol{b}$ is affine, or linear on the homogeneous coordinates $(\boldsymbol{x}, 1)$. Its flow can be analytically solved

$$\begin{bmatrix} \Phi_C^\tau(\boldsymbol{x}) \\ 1 \end{bmatrix} = \exp\left(\tau \begin{bmatrix} J\boldsymbol{A} & J\boldsymbol{b} \\ \boldsymbol{0}^T & 0 \end{bmatrix}\right) \begin{bmatrix} \boldsymbol{x} \\ 1 \end{bmatrix} \tag{4}$$

using the matrix exponential $\exp(\cdot)$ for which efficient numerical algorithms exist [Moler and Van Loan, 2003]. This equation can be verified to have the correct scalar field and boundary condition, and thus forms the unique solution to the ODE in Equation (3).

## 3.2 Symmetrising observables

Given an observable $C \in \mathcal{O}$, we want to transform an observable $f$ into $\hat{f}$ that is invariant to the transformations generated by $C$. This means that $\hat{f} \circ \Phi_C^\tau = \hat{f}$ for all symmetry time $\tau \in \mathbb{R}$. Via Noether's theorem, we know that this is equivalent to $C$ being conserved in the trajectories generated by $f$, and also equivalent to $\{C, \hat{f}\} = 0$. However, this equation does not prescribe how to obtain such $\hat{f}$. Instead, we'll create $\hat{f}$ by symmetrizing over the symmetry group generated by $C$. This is done by averaging over the orbit of the transformation

$$\hat{f}(\boldsymbol{x}) = \int_{\mathbb{R}} f(\Phi_C^\tau(\boldsymbol{x})) \mu(\tau).$$

with a measure $\mu$ over symmetry time $\tau$. This measure $\mu$ induces a measure on the 1-dimensional subgroup $\mathcal{G}_C$ of the group of diffeomorphisms $\mathcal{M} \to \mathcal{M}$. If this measure on $\mathcal{G}_C$ is uniform (specifically, a right-invariant measure [Halmos, 1950]), then $\hat{f}$ is indeed invariant.

Instead of a single symmetry generator, we can also have a set $\mathcal{C} = \{C_1, ..., C_K\}$ of observables and we want to make $f$ invariant to all of these. Assume that this set spans a vector space of observables that is closed under the Poisson bracket (i.e. they form a Lie subalgebra). In that case, the groups of transformations of the observables combined generate a group $\mathcal{G}_C$ [Hall, 2015, Thm. 5.20]. This group is parameterized by a vector of symmetry times $\boldsymbol{\tau} \in \mathbb{R}^K$. The corresponding flow is $\Phi_C^\tau = \Phi_{\sum_i \tau_i C_i}^1$. To make an observable $f$ invariant to the symmetries of all conserverved quantities $\mathcal{C}$, equivalently to the group $\mathcal{G}_C$, we symmetrize

$$\hat{f}(\boldsymbol{x}) = \int_{\mathbb{R}^K} f(\Phi_C^\tau(\boldsymbol{x})) \mu(\boldsymbol{\tau}). \tag{5}$$

with some measure $\mu$ over $\mathbb{R}^K$. As before, if this induces a uniform measure over $\mathcal{G}_C$, then this symmetrization indeed makes $\hat{f}$ invariant to $\mathcal{G}_C$.

However, a probability measure $\mu(\boldsymbol{\tau})$ that gives a uniform distribution over $\mathcal{G}_C$ might not exist, for example when the group contains a non-compact group of translations. Even when such a measure does exist, it may be hard to construct, and the symmetrisation integral in Equation (5) may be intractable to compute. So instead, in practice, we approximate this by choosing $\mu(\boldsymbol{\tau})$ to be a unit normal distribution $\mathcal{N}(0, \boldsymbol{I}_K)$ or uniform distribution. This results in a relaxed notion of symmetry

in $\hat{f}$ which can be interpreted as a form of robustness to actions of the symmetry group implied by the conserved quantity, by smoothing the function in this direction around data, in contrast to strict invariance by definition closed under group actions along the full orbit. Finally, we approximate the integral by an unbiased Monte Carlo estimate with $S$ samples.

# 4 Automatic Symmetry Discovery using Noether's Razor

Now that we have a way of parameterising symmetry differentiably as conservation laws through Noether's theorem, we need an objective function that is capable of selecting the right symmetry. Unfortunately, regular training objectives that only rely on data fit can not necessarily distinguish the correct inductive bias, as noted in prior work [van der Wilk et al., 2018, Immer et al., 2022, van der Ouderaa and van der Wilk, 2021]. This is because, even if train data originates from a symmetric distribution, there can be both non-symmetric and symmetric solutions that fit the train data equally well, given a sufficiently flexible model. Consequently, the regular maximum likelihood objective that only measures train data fit will not necessarily favour a symmetric model, even if we expect this to generalise best on test data. Instead of having to resort to cross-validation to select the right symmetry inductive bias, we propose to use an approximate marginal likelihood on the train data. This has the additional benefit of being differentiable, allowing symmetrisation to be learned with back-propagation along with regular parameters in a single training procedure. In our case, we use *Noether's theorem* to parameterise symmetries in our prior through conserved quantities, which we can optimise with back-propagation using a differentiable lower bound on the marginal likelihood. This quantity, also known as the 'evidence', differs distinctly from maximum likelihood in that it balances both train fit as well as model complexity. The *Occam's razor* effect encourages symmetry and leverages the symmetrisation process to 'cut away' prior density over Hamiltonian functions that are not symmetric, if this does not result in a worse data fit. The resulting posterior predictions automatically becomes symmetric if observed data obeys a symmetry (high evidence for symmetry), but can become non-symmetric if this does not match the data (low evidence for symmetry). Hence, the name of our proposed method for automatic inductive bias selection is *Noether's razor*.

## 4.1 Probabilistic model with symmetries embedded in the prior.

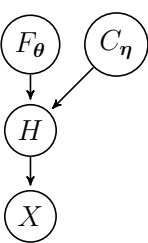

To be more explicit about our probabilistic model, we can introduce four variables, namely a non-symmetrised observable $F_{\boldsymbol{\theta}}$, a set of conserved quantities $C_{\boldsymbol{\eta}}$, which induce a symmetrised Hamiltonian $H$ generating the observed trajectory data $X$. We treat trajectory data an an observed variable, consider the conserved quantities as part of an empirical prior as we optimise over them, and integrate out the Hamiltonian as latent. The construction can be interpreted a placing a sophisticated prior over the functions that the symmetrised Hamiltonian $H$ can represent, which is the variable of primary interest. The underlying non-symmetrised $F_{\boldsymbol{\theta}}$ does not have a direct physical meaning as $H$ does, but defines a prior over neural networks to flexibly define a density over a rich class of possible functions. The conserved quantities $C_{\boldsymbol{\eta}}$ control the amount of symmetry in the effective prior over symmetrised Hamiltonians $H$. Empirically optimising $C_{\boldsymbol{\eta}}$ through Bayesian model selection allows us to 'cut

Figure 1: Graphical probabilistic model. Trajectory data $X$ depends on a symmetrised Hamiltonian $H$ induced by non-symmetrised observable $F$ and conservation laws $C$.

away' density in the prior over $H$ that correspond to functions that are not symmetric - as the symmetrisation averages functions in $F_{\boldsymbol{\theta}}$ that lie in the same orbit and thereby increases the relative density of symmetric functions in $H$. We hypothesise that we will not over-fit conserved quantities as $\boldsymbol{\eta}$ is relatively low-dimensional, only representing quadratic functions, while we integrate out the high-dimensional neural network model parameters $\boldsymbol{\theta}$ that parameterises the observable $F_{\boldsymbol{\theta}}$. In future work, it would be interesting to explore a richer function classes for conserved quantities, such as neural networks, although we do expect this to be more difficult and to require additional priors or regularisation techniques to avoid over-fitting.

## 4.2 Bayesian model selection for symmetry discovery

To learn the right symmetry from data, we propose to use Bayesian model selection through optimisation of the marginal likelihood. In the previous sections, we have phrased symmetries parameterised by $\boldsymbol{\eta}$ as part of the prior over Hamiltonians. The symmetry parameters $\boldsymbol{\eta}$ parameterise the space of possible 'models' that we consider, whereas the model parameters $\boldsymbol{\theta}$ parameterise the weights of a single model. To perform Bayesian model selection on the symmetries, we are interested in computing the marginal likelihood:

$$p(\boldsymbol{x}|\boldsymbol{\eta}) = \int_{\boldsymbol{\theta}} p(\boldsymbol{x}|\boldsymbol{\theta},\boldsymbol{\eta})p(\boldsymbol{\theta})\mathrm{d}\boldsymbol{\theta} \tag{6}$$

which requires integrating (marginalising) the likelihood over model parameters $\boldsymbol{\theta}$ weighted by the prior, and is sometimes referred to as the 'evidence' for a particular model. Unlike maximum likelihood, the marginal likelihood has an Occam's razor effect [Smith and Spiegelhalter, 1980, Rasmussen and Ghahramani, 2000] that balancing both data fit and model complexity, allowing optimisation of symmetry parameters $\boldsymbol{\eta}$. Although the marginal likelihood is typically intractable, certain approximate Bayesian inference techniques can provide differentiable estimates. In the next sections, we will use variational inference to derive a tractable and differentiable lower bound to the marginal likelihood that can be used to find a posterior over $\boldsymbol{\theta}$ and optimise symmetries $\boldsymbol{\eta}$.

**Why the marginal likelihood can learn symmetry** To understand why the marginal likelihood objective is capable of learning the right symmetry (to learn $\boldsymbol{\eta}$), Sec. 3.2 [van der Wilk et al., 2018] proposed to decompose it through the product rule:

$$p(\boldsymbol{x} \mid \boldsymbol{\eta}) = p(\boldsymbol{x}_1 \mid \boldsymbol{\eta})p(\boldsymbol{x}_2|\boldsymbol{x}_1,\boldsymbol{\eta})p(\boldsymbol{x}_3 \mid \boldsymbol{x}_{1:2},\boldsymbol{\eta}) \prod_{c=4}^{C} p(\boldsymbol{x}_c \mid \boldsymbol{x}_{1:c-1},\boldsymbol{\eta}) \tag{7}$$

which shows that the marginal likelihood measures how much parts of the dataset predict other parts of the data - a measure of generalisation that does not require cross-validation. Given a perfect data fit, the marginal likelihood will be higher when the right symmetry is selected, as parts of the dataset will result in better and more certain predictions on other part of the data. This is unlike the maximum likelihood, which is always maximised with perfect data fit, with or without the right symmetry. For some posterior approximations, such as linearised Laplace approximations, it can be analytically shown that symmetry maximises the approximate marginal likelihood (App. G.2 of Immer et al. [2022]). Our method is very similar, but uses more expressive variational inference which can optimise the posterior globally, rather than relying on a local Taylor expansion.

## 4.3 Lower bounding the marginal likelihood

The marginal likelihood of an Hamiltonian neural network is typically not tractable in closed-form. However, we can derive a lower bound to the marginal likelihood using variational inference (VI):

$$\log p(\boldsymbol{x} \mid \boldsymbol{\eta}) \geq \mathbb{E}_{\boldsymbol{\theta}} \left[\log p(\boldsymbol{x} \mid \boldsymbol{\theta},\boldsymbol{\eta})\right] - \mathrm{KL}(q_{\boldsymbol{m},\boldsymbol{S}}(\boldsymbol{\theta}) \mid\mid p(\boldsymbol{\theta})) \tag{8}$$

$$\geq \mathbb{E}_{\boldsymbol{\theta}} \left[\mathbb{E}_{\boldsymbol{\tau}} \left[\sum_{i=1}^{N} \log \mathcal{N}(\boldsymbol{x}_{t'}^i \mid \widehat{H}_{\boldsymbol{\theta},\boldsymbol{\eta}}^{\boldsymbol{\tau}}(\boldsymbol{x}_t^i), \sigma_{\mathrm{data}}^2 \boldsymbol{I})\right]\right] - \mathrm{KL}(q_{\boldsymbol{m},\boldsymbol{S}}(\boldsymbol{\theta}) \mid\mid p(\boldsymbol{\theta} \mid \boldsymbol{0}, \sigma_{\mathrm{prior}}^2 \boldsymbol{I}))$$

where $\widehat{H}_{\boldsymbol{\theta},\boldsymbol{\eta}}^{\boldsymbol{\tau}}(\boldsymbol{x}_t^i) = \frac{1}{S}\sum_{s=1}^{S} H_{\boldsymbol{\theta},\boldsymbol{\eta}}(\boldsymbol{\Phi}_{\boldsymbol{\eta}}^{\boldsymbol{\tau}^{(s)}}(\boldsymbol{x}_t^i))$ and $\widehat{H}$ is an unbiased $S$-sample Monte Carlo estimator of the symmetrised Hamiltonian. We write $\mathbb{E}_{\boldsymbol{\theta}} := \mathbb{E}_{\boldsymbol{\theta}\sim q_{\boldsymbol{m},\boldsymbol{S}}}$ and $\mathbb{E}_{\boldsymbol{\tau}} = \mathbb{E}_{\boldsymbol{\tau}\sim \prod_{s=1}^{S}\mu(\boldsymbol{\tau})}$ for which we can obtain an unbiased estimate by taking Monte Carlo samples. The first inequality is the standard VI lower bound. The second inequality follows from applying Jensen's inequality (again) which uses the fact that the log likelihood is a convex function. Similar lower bounds to invariant models that average over a symmetry group have recently appeared in prior work [van der Ouderaa and van der Wilk, 2021, Schwöbel et al., 2022, Nabarro et al., 2022]. Full derivation in Appendix A.1.

## 4.4 Improved variational inference for scalable Bayesian model selection

Variational inference is a common tool to perform Bayesian inference on models with intractable marginal likelihoods, including neural networks. In deep learning literature, however, its use is typically limited to better predictive uncertainty estimation and rarely for Bayesian model selection. Meanwhile, linearised Laplace approximations have recently been successfully applied to Bayesian

model selection [Immer et al., 2021] and symmetry learning in specific [Immer et al., 2022, van der Ouderaa et al., 2024], with a few reported cases of model selection using VI only in single neural network layers [van der Ouderaa and van der Wilk, 2021, Schwöbel et al., 2021]. Optimising Bayesian neural networks with variational inference is much less established than training regular neural networks, for which many useful heuristics are available. This work, however, provides evidence that it is also possible to perform approximate Bayesian model selection using VI in deep neural networks, which we deem an interesting observation in its own right. To make sure the lower bound on the marginal likelihood is sufficiently tight, we employ a series of techniques, including a richer non-mean field family of matrix normal posteriors [Louizos and Welling, 2016], and closed-form updates of the prior precision and output variances derived with expectation maximisation. Details on how we train a Bayesian neural network using variational inference can be found in Appendix D.

## 5  Results

In this section, we will discuss how the learned symmetries are analysed and then list our experiments and results.

### 5.1  Analyzing learned symmetries

In our experiments, we will find a set of $K$ conserved quantities $C_k : \mathcal{M} \to \mathbb{R}$. As we consider quadratic conserved quantities in particular, we can equivalently analyze the resulting generators of the associated symmetries $\hat{G}_k(\boldsymbol{x}) = J\nabla C_k$ which are affine and thus representable with a matrix $\hat{G}_k \in \mathbb{R}^{(M+1)\times(M+1)}$ on homogeneous coordinates $(\boldsymbol{x}, 1)$. In Appendix B, we list for each system the $L$ ground truth conserved quantities generators $G_l^\star$. The learned and ground truth generators can be stacked in to the matrices $\hat{\boldsymbol{G}} \in \mathbb{R}^{K\times(M+1)^2}, \boldsymbol{G}^* \in \mathbb{R}^{L\times(M+1)^2}$ respectively. As we can identify the symmetries only up to linear combinations, we have learned the correct symmetries if the learned generators span a linear subspace of $\mathbb{R}^{(M+1)^2}$ that coincides with the space spanned by the ground truth generators. To verify this, we test two properties. First, we show that the matrix $\hat{\boldsymbol{G}}$ has $L$ non-zero singular values. Secondly, for the first $L$ right singular vectors $v_i \in \mathbb{R}^{(M+1)^2}$, we decompose $v_i = v_i^\parallel + v_i^\perp$ in a vector in ground truth subspace, and one orthogonal to it. The learned $v_i$ is a correct conserved quantity if $v_i^\perp = 0$, or equivalently, because the singular vectors are normalized, if $\|v_i^\parallel\| = 1$. We call this measure the "parallelness".

### 5.2  Simple Harmonic Oscillator

We start with a demonstration on the simple harmonic oscillator. This text book example has a 2-dimensional phase space, making learned Hamiltonians amenable to visualisation. Further, it has a clear rotational symmetry SO(2), relating to the conserved phase. On a finite set of generated train data, we model the Hamiltonian using a vanilla HNN, our symmetry learning

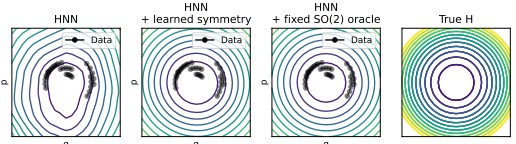

Figure 2: Learned Hamiltonians on phase space of simple harmonic oscillator by HNN models.

method, and a model with true symmetry built-in as reference oracle (experimental details in Appendix B.1). In Figure 2, we find that our symmetry learning method results in a rotationally invariant Hamiltonian that matches the fixed rotational SO(2) symmetry. Further away from the origin, the learned Hamiltonian differs from the ground truth Hamiltonian, as there is no data in that region. In Table 1, we find that the learned symmetry has a better ELBO on the train set and matches the improved predictive performance of the model with the correct symmetry built-in. The symmetry learning method outperforms the vanilla model in terms of predictive performance on the test set.

Table 1: Learning Hamiltonian dynamics of the simple harmonic oscillator. We compare a vanilla HNN, our symmetry learning method, and a model with the correct SO(2) symmetry built-in as reference oracle. Our method achieves reference oracle performance, indicating correct symmetry learning, and outperforms the vanilla model by improving predictive performance on the test set.

| Learned dynamics: **simple harmonic oscillator** | | Train data | | | | Test data |
|---|---|---|---|---|---|---|
| | | Train MSE | NLL/N | KL/N | -ELBO/N ($\downarrow$) | Test MSE ($\downarrow$) |
| HNN | | 0.005 | 0.3667 | 3314.374 | 3314.741 | 0.005 |
| HNN + learned symmetry | (**ours**) | 0.002 | -2.618 | 3304.754 | **3302.136** | **0.002** |
| HNN + fixed SO(2) | (reference oracle) | 0.002 | -3.213 | 3298.357 | **3295.144** | **0.002** |

## 5.3  $n-$Harmonic Oscillators

Now, we consider $n-$harmonic oscillators. This system has as symmetry group the unitary Lie group U($n$) of dimensionality of $n^2$ (see Appendix B.2). We sample random trajectories from phase space and train a HNN neural network without and with symmetry learning using variational inference. Again, we find improved ELBO and test performance for learned symmetries Table 2. Following the protocol from Section 5.1, we analyze the learned symmetries. In Figure 3 (right), we see that for varying $n$, we indeed find that the matrix of learned symmetries has

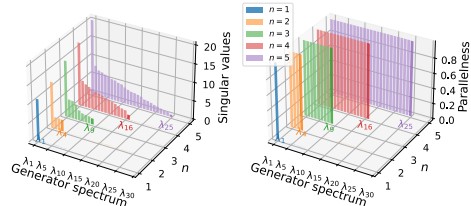

Figure 3: Singular value and parallelness of the singular vectors of the learned generators, for $n$ oscillators. U($n$) is correctly learned.

$n^2$ nonzero singular values. Furthermore, the first $n^2$ singular vectors lie in the ground truth subspace of generators with measured parallelness $\|v_i^\parallel\| > 0.99$, as seen in Figure 3 (left). This shows that the $U(n)$ symmetry is corectly learned.

Table 2: Learning Hamiltonian dynamics of $3-$fold harmonic oscillators. We compare HNN with symmetry learning to a vanilla HNN without symmetry learning and to the correct U(3) symmetry built-in as fixed reference oracle. We find that our method can discover the correct symmetry, achieves reference oracle performance, and outperforms vanilla training in both ELBO and test performance.

| Learned dynamics: **simple harmonic oscillator** | | Train data | | | | Test data |
|---|---|---|---|---|---|---|
| | | Train MSE | NLL/N | KL/N | -ELBO/N ($\downarrow$) | Test MSE ($\downarrow$) |
| HNN | | 0.00106 | -12.04 | 5.27 | -6.77 | 0.00002141 |
| HNN + learned symmetry | (**ours**) | 0.00102 | -12.16 | 2.53 | **-9.63** | **0.00000994** |
| HNN + fixed symmetry U($n$) | (reference oracle) | 0.00102 | -12.15 | 2.21 | **-9.94** | **0.00000898** |

## 5.4  $n$-Body System

To investigate performance of our method on more interesting systems, we consider learning the Hamiltonian of an $n$-body system with gravitational interaction. We use 3 bodies in 2 dimensions so that trajectories and generators remain easy to visualise. As the Hamiltonian depends only on the norm of the momenta and on positions via the relative distances of the bodies, the three dimensional group SE(2) of rototranslations is an invariance of the ground truth Hamiltonian. However, as explained in Appendix B.3, the Hamiltonian has four more quadratic conserved quantities. They generate a 7-dimensional

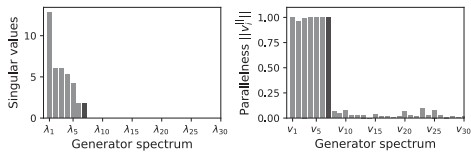

Figure 4: Singular value and parallelness of the singular vectors of the learned generators for three body system in two dimensions. The 7-dimensional Lie group $\mathcal{G}$ of quadratic conserved quantities is correctly learned.

Lie group $\mathcal{G}$ of symmetries. This group has the same orbits on the phase space as SE(2). Therefore, a function being invariant to SE(2) is equivalent to it being invariant to $\mathcal{G}$. We'll find that Noether's razor discovers not just SE(2), but all seven generators of $\mathcal{G}$.

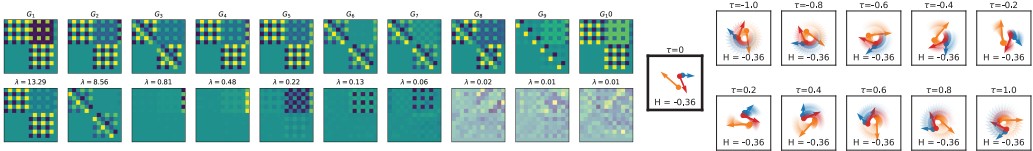

Figure 5: Learned generators associated by conserved quantities and their singular value decomposition. We find a subspace spanned by the 7 linear generators that correspond to the correct symmetries (see Appendix B.3): (1) rotation of the center of mass $R^{\mathrm{COM}}$, (2) rotation around the origin $R^{\mathrm{ABS}}$, (4+5) translation, (5+6+7) momentum-dependent translations $P, Q$, (8+9+10) inactive ($\lambda < 0.05$). The first 7 singular vectors lie in the ground truth subspace of generators with measured parallelness $||v_i^{||}|| > 0.95$.

In Table 3, we compare performance of a vanilla variational HNN with our symmetry learning approach and a model that has the appropriate SE(2) symmetry of rototranslations built-in as an reference oracle. We find that our method is able to *automatically discover* the conserved quantities and associated generators that span the symmetry group. The model achieves the same performance as the model with the symmetry built-in as reference oracle, but without having required the prior knowledge. Compared to the vanilla baseline, our approach improves test accuracy on both in-distribution as out-of-distribution test sets.

Table 3: Learning Hamiltonian dynamics of 2d 3-body system with variational Hamiltonian neural networks (HNN). We compare our symmetry learning method to a vanilla model without symmetry learning and a model with the correct SE(2) symmetry built-in as a reference oracle. Our method capable of discoverying symmetry achieves the oracle performance, outperforming the vanilla method.

| Learned dynamics:
**2d 3-body system** | | Train data | | | | Test data | Test data (moved) | Test data (wider) |
| --- | --- | --- | --- | --- | --- | --- | --- | --- |
| | | Train MSE | NLL/N | KL/N | -ELBO/N ($\downarrow$) | Test MSE ($\downarrow$) | Test MSE ($\downarrow$) | Test MSE ($\downarrow$) |
| HNN | | 0.0028 | -13.87 | 13.34 | -9.52 | 0.0016 | 0.0035 | 0.0016 |
| HNN + learned symmetry | **(ours)** | 0.0017 | -20.09 | 7.28 | **-12.81** | **0.0006** | **0.0004** | **0.0006** |
| HNN + fixed SE(2) | (reference oracle) | 0.0019 | -19.27 | 7.96 | **-11.32** | **0.0006** | **0.0006** | **0.0006** |

After training, we can analyse the learned conserved quantities and implied symmetries by inspecting their associated generators. In Figure 5, we plot these generators as well as their singular value decomposition. We see that our method correctly learns 7 singular values with $\lambda_i > 0.05$ and the associated singular vectors lie in the ground truth subspace with $\|v_i^{||}\| > 0.95$. This indicates that our method is in fact capable of inferring the right symmetries from train data, beyond merely improving generalisation by improving predictive performance on the test set.

## 6 Conclusion

In this work, we propose to use *Noether's theorem* to parameterise symmetries in machine learning models of dynamical systems in terms of conserved quantities. Secondly, we propose to leverage the *Occam's razor* effect of Bayesian model selection by phrasing symmetries implied by conserved quantities in the prior and learning them by optimising an approximate marginal likelihood directly on train data, which does not require validation data or explicit regularisation of the conserved quantities. Our approach, dubbed *Noether's razor*, encourages symmetries by balancing both data fit and model complexity. We derive a variational lower bound on the marginal likelihood providing a concrete objective capable of jointly learning the neural network as well as the conserved quantities that symmetrise the Hamiltonian. As far as we know, this is also the first time differentiable Bayesian model selection using variational inference has been demonstrated on deep neural networks. We demonstrate our approach on $n$-harmonic oscillators and $n$-body systems. We find that our method learns the correct conserved quantities by analysing the singular values and correctness of the subspace spanned by the generators implied by learned conserved quantitites. Further, we find that our method performs on-par with models with the true symmetries built-in explicitly and we outperform vanilla model, improving generalisation and predictive accuracies on test data.

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

# A  Mathematical derivations

## A.1  ELBO of Hamiltonian Neural Network

We can find a lower bound on the marginal likelihood $\log p(\boldsymbol{x} \mid \boldsymbol{\eta})$ through variational inference,

$$\log p(\boldsymbol{x} \mid \boldsymbol{\eta}) \geq \mathbb{E}_{\boldsymbol{\theta}}\left[\log p(\boldsymbol{x} \mid \boldsymbol{\theta}, \boldsymbol{\eta})\right] - \mathrm{KL}(q_{\boldsymbol{m},\boldsymbol{S}}(\boldsymbol{\theta}) \parallel p(\boldsymbol{\theta})) \tag{9}$$

$$= \mathbb{E}_{\boldsymbol{\theta}}\left[\sum_{i=1}^{N} \log \mathcal{N}(\boldsymbol{x}_{t'}^i \mid H_{\boldsymbol{\theta},\boldsymbol{\eta}}(\boldsymbol{x}_t^i), \sigma_{\mathrm{data}}^2 \boldsymbol{I})\right] - \mathrm{KL}(q_{\boldsymbol{m},\boldsymbol{S}}(\boldsymbol{\theta}) \parallel p(\boldsymbol{\theta} \mid \boldsymbol{0}, \sigma_{\mathrm{prior}}^2 \boldsymbol{I})) \tag{10}$$

$$= \mathbb{E}_{\boldsymbol{\theta}}\left[\sum_{i=1}^{N} \log \mathcal{N}(\boldsymbol{x}_{t'}^i \mid \mathbb{E}_{\boldsymbol{\tau}}\left[\widehat{H}_{\boldsymbol{\theta},\boldsymbol{\eta}}^{\boldsymbol{\tau}}(\boldsymbol{x}_t^i)\right], \sigma_{\mathrm{data}}^2 \boldsymbol{I})\right] - \mathrm{KL}(q_{\boldsymbol{m},\boldsymbol{S}}(\boldsymbol{\theta}) \parallel p(\boldsymbol{\theta} \mid \boldsymbol{0}, \sigma_{\mathrm{prior}}^2 \boldsymbol{I})) \tag{11}$$

$$\geq \mathbb{E}_{\boldsymbol{\theta}}\left[\mathbb{E}_{\boldsymbol{\tau}}\left[\sum_{i=1}^{N} \log \mathcal{N}(\boldsymbol{x}_{t'}^i \mid \widehat{H}_{\boldsymbol{\theta},\boldsymbol{\eta}}^{\boldsymbol{\tau}}(\boldsymbol{x}_t^i), \sigma_{\mathrm{data}}^2 \boldsymbol{I})\right]\right] - \mathrm{KL}(q_{\boldsymbol{m},\boldsymbol{S}}(\boldsymbol{\theta}) \parallel p(\boldsymbol{\theta} \mid \boldsymbol{0}, \sigma_{\mathrm{prior}}^2 \boldsymbol{I})) \tag{12}$$

$$\approx \frac{1}{M}\sum_{i=1}^{M}\left[\sum_{i=1}^{N} \log \mathcal{N}(\boldsymbol{x}_{t'}^i \mid \frac{1}{S}\sum_{s=1}^{S} H_{\boldsymbol{\theta},\boldsymbol{\eta}}(\boldsymbol{\Phi}_{\mathcal{C}_{\boldsymbol{\eta}}}^{\boldsymbol{\tau}^{(s)}}(\boldsymbol{x}_t^i)), \sigma_{\mathrm{data}}^2 \boldsymbol{I})\right] - \mathrm{KL}(q_{\boldsymbol{m},\boldsymbol{S}}(\boldsymbol{\theta}) \parallel p(\boldsymbol{\theta} \mid \boldsymbol{0}, \sigma_{\mathrm{prior}}^2 \boldsymbol{I})) \tag{13}$$

where we use $M$ samples to obtain an unbiased estimate of $\mathbb{E}_{\boldsymbol{\theta}} := \mathbb{E}_{\boldsymbol{\theta} \sim q_{\boldsymbol{m},\boldsymbol{S}}}$ and a single sample $\mathbb{E}_{\boldsymbol{\tau}} = \mathbb{E}_{\boldsymbol{\tau} \sim \prod_{s=1}^{S} p(\boldsymbol{\tau})}$ and use an $S$-sampled Monte Carlo estimate of the symmetrised Hamiltonian:

$$\widehat{H}_{\boldsymbol{\theta},\boldsymbol{\eta}}(\boldsymbol{x}_t^i) = \frac{1}{S}\sum_{s=1}^{S} H_{\boldsymbol{\theta},\boldsymbol{\eta}}(\boldsymbol{\Phi}_{\mathcal{C}_{\boldsymbol{\eta}}}^{\boldsymbol{\tau}^{(s)}}(\boldsymbol{x}_t^i)) \text{ with samples } \boldsymbol{\tau}^{(1)}, \boldsymbol{\tau}^{(2)}, \ldots, \boldsymbol{\tau}^{(S)} \sim \mu(\boldsymbol{\tau}) \tag{14}$$

with the fact that this yields an unbiased estimator of the true symmetrised Hamiltonian $\mathbb{E}_{\boldsymbol{\tau}}\left[\widehat{H}_{\boldsymbol{\theta},\boldsymbol{\eta}}(\cdot)\right] = \widehat{H}_{\boldsymbol{\theta},\boldsymbol{\eta}}(\cdot)$. where we obtained an unbiased estimate of expectations through $S$ sampled symmetry transformations and $M$ sampled parameters. The first inequality is the standard VI lower bound. The second inequality follows from applying Jensen's inequality (again), using the fact that the log likelihood is convex. Similar bounds to symmetrisation by averaging over orbits have appeared in prior work [van der Ouderaa and van der Wilk, 2021, Schwöbel et al., 2022, Nabarro et al., 2022].

# B  Ground truth conserved quantities

In this section, we'll discuss the conserved quantities present in the ground-truth Hamiltonians of the systems we discuss.

As stated in Section 2.3, we can combine conserved quantities into new ones by linear combinations, products, and Poisson brackets. Thus we'll speak of the generating set of conserved quantities, which combine into all conserved quantities.

## B.1  Simple harmonic oscillator

For the simple harmonic oscillator, the phase space is $\mathbb{R} \times \mathbb{R}$. The ground truth Hamiltonian is

$$H(p, q) = \frac{p^2}{2m} + \frac{kq^2}{2}.$$

We choose $m = k = 1$, so that $H(p, q) = (p^2 + q^2)/2$. Time evolution is a rotation of phase space. The Hamiltonian itself generates all conserved quantities.

## B.2   $n-$**Simple harmonic oscillators**

The phase space is $\mathbb{R}^{2n}$. We choose all $k = m = 1$, so that the Hamiltonian is

$$H(p,q) = \|p\|^2/2 + \|q\|^2/2.$$

Time evolution rotates each pair $(q_i, p_i)$. The conserved quantities are generated by the following set, for $i, j = 1, ..., n, i \neq j$.

$$H_i = (q_i^2 + p_i^2)/2$$
$$R_{ij} = q_i p_j - q_j p_i$$
$$F_{ij} = q_i q_j + p_i p_j$$

The conserved quantity $H_i$ rotates the pair $(q_i, p_i)$. $R_{ij}$ rotates both pairs $(q_i, q_j)$ and $(p_i, p_j)$ and $F_{ij}$ rotates both pairs $(q_i, p_j)$ and $(q_j, p_i)$.

Alternatively, we can interpret the phase space as $\mathbb{C}^n$ [Arnold, 1989, Sec. 41E], with the positions being the real part and the momenta the imaginary part. In that case, the symplectic form $J$ becomes simply the complex number $-i$. Then $H(\boldsymbol{x}) = \boldsymbol{x}^\dagger \boldsymbol{x}/2$. Time evolution is multiplication by the complex number $e^{-it}$. Conserved quantities are $H_i = x_i^* x_i/2$, which is real, and $C_{ij} = x_i^* x_j$, whose real part corresponds to $F_{ij}$ and imaginary part to $R_{ij}$. The conserved quantities $H_i$ and $C_{ij}$ are quadratic and thus their symmetries are generated by linear matrices, which are all skew-Hermitian. In fact, all skew-Hermitian matrices are spanned by these generators. This shows that the combined symmetry group is in fact $U(n)$ [Amiet and Weigert, 2002].

## B.3   $n$-**body**

In $D$ spatial dimensions, with $n$ bodies, the phase space is $\mathbb{R}^{2nD}$ and the Hamiltonian is

$$H(p,q) = \sum_i \frac{\|p_i\|^2}{2m_i} + \sum_{i \neq j} \frac{Gm_i m_j}{\sqrt{\|q_i - q_j\|^2 + \epsilon^2}}$$

with a small $\epsilon$ to make it smooth.

The main conserved quantities are, for $d, d' = 1, ...D, d \neq d'$,

$$T_d = \sum_i p_{id}$$

$$R_{dd'}^{\text{ABS}} = \sum_i (q_i \wedge p_i)_{dd'}$$

$$R_{dd'}^{\text{COM}} = (q_{\text{COM}} \wedge p_{\text{COM}})_{dd'}$$

with $q_{\text{COM}} = \sum_i m_i q_i / \sum_i m_i$ and $p_{\text{COM}} = \sum_i m_i p_i / \sum_i m_i$.

These generate further conserved quantities of interest:

$$P_d = T_d^2$$
$$Q_{dd'} = T_d T_{d'}$$
$$R_{dd'}^{\text{REL}} = \sum_i ((q_i - q_{\text{COM}}) \wedge (p_i - p_{\text{COM}})_{dd'} = R_{dd'}^{\text{ABS}} - 2R_{dd'}^{\text{COM}}$$

As $T_d$ is linear, we can still learn $P_d$ and $Q_{dd'}$ as quadratic conserved quantities. As the $R^{\text{REL}}$ is a linear combination of other conserved quantities, we disregard it in our analysis of the learned symmetries.

The corresponding symmetries are: $T_d$ translates all bodies in the $d$ direction. $R_{dd'}^{\text{COM}}$ rotates the center of mass of all the bodies in the plane $dd'$, while preserving the positions relative to the center of mass. $R^{\text{ABS}}$ rotates all bodies relative to the origin. $R^{\text{REL}}$ rotates all bodies relative to the center of mass. $P_d$ translates in the $d$ direction proportional to its COM momentum. $Q_{dd'}$ translates in direction $d$ proportional to $p_{\text{COM},d'}$ and vice versa.

These symmetries together generate a group we'll call $\mathcal{G}$. This group has as a subgroup SE($D$), which is generated by $T$ and $R^{\mathrm{ABS}}$. The group $\mathcal{G}$ has the same orbits as SE($D$), as each element in $\mathcal{G}$ can be seen as a rototranslation conditional on some property of phase space. Because the orbits are the same, for any observable $f : \mathcal{M} \to \mathbb{R}$, we have that $f$ invariant to $\mathcal{G}$ is true if and only if $f$ is invariant to SE(2).

This system can have further conserved quantities, such as the Laplace–Runge–Lenz vector for $n = 2$. However, these are not expressible as a quadratic polynomial. As far as we know, the conserved quantities listed above are all that are expressible as a quadratic polynomial.

## C   Experimental details

All experiments were run on a single NVIDIA RTX 4090 GPU with 24GiB of GPU memory.

### C.1   Simple harmonic oscillator experiment

**Data**   For the training data, we sampled 7 initial conditions from unit Gaussian and simulated 4 datapoints with $\Delta t = 0.2$ apart. For test data, we sampled 100 initial conditions from unit Gaussian and simulated 20 timesteps with $\Delta t = 0.2$ from each initial condition.

**Training**   We use an MLP with 2 hidden layers, each consisting of 200 hidden neurons and a linear exponential unit activation function with $\alpha = 2$. For symmetrisation, we use $S = 200$ samples from a uniform measure for $\mu(\tau)$. We use 20 Euler steps for time integration. We use fixed output noise and closed-form prior variance (Appendix D). We optimise the ELBO in full batch with Adam [Kingma and Ba, 2014] ($\beta_1 = 0.9, \beta_2 = 0.999$) trained for 2000 epochs with a learning rate of 0.001, cosine annealed to 0.

### C.2   $n-$Simple harmonic oscillators experiment

**Data**   For training data, we randomly sampled 200 initial conditions independently from a unit normal. For each initial condition, we simulated a trajectory consisting of 50 data points at 0.3 time units apart.

**Training**   We use an MLP with 3 hidden layers with 200 hidden units and exponential linear activation functions with $\alpha = 1$. We optimise the ELBO in mini batches of $B = 20$ trajectories, using $S = 100$ symmetrisation samples, 20 Euler steps for time integration, and $M = 2$ weight samples using Adam [Kingma and Ba, 2014] ($\beta_1 = 0.9, \beta_2 = 0.999$) for 2000 epochs with a learning rate starting from 0.001, cosine annealed to 0.

### C.3   $n$-body experiment

**Data**   For training data, we randomly sampled 200 initial conditions by independently sampling positions from a unit normal, shifted by a normal with a standard deviation of 3. From each initial condition, we simulated trajectories consisting of 50 data points 0.3 time units apart.

**Training**   We use an MLP with 4 hidden layers with 250 hidden unit units and exponential linear unit activation functions with $\alpha = 1$. We optimise the ELBO batches of $B = 20$ trajectories, $S = 100$ symmetrisation samples, 20 Euler steps for time integration, and $M = 2$ weight samples using Adam [Kingma and Ba, 2014] ($\beta_1 = 0.9, \beta_2 = 0.999$) for 2000 epochs with a learning rate starting from 0.001, cosine annealed to 0.

# D  On training a neural network with variational inference

## D.1  Matrix normal variational posterior

Naively, the covariance of a Gaussian posterior over weights grows quadratically with the number of parameters $|\boldsymbol{\theta}|^2$. It is therefore common to disregard all correlations between weights, resulting in a diagonal or mean-field posterior. Although the ELBO remains a lower bound for any choice of approximate family, more crude approximations can increase the slack in the bound, possibly making it harder to use estimates for Bayesian model selection. We, therefore, propose to use matrix normal posteriors [Louizos and Welling, 2016] factorised per layer,

$$q(\boldsymbol{\theta}) = \prod_{i=1}^{N} q(\boldsymbol{\theta}_l), \text{ with} \qquad q(\boldsymbol{\theta}_l) = \mathcal{N}(\boldsymbol{\theta}_l \mid, \boldsymbol{m}_l, \boldsymbol{S}_l \otimes \boldsymbol{A}_l) \qquad (15)$$

where $\boldsymbol{\theta} = (\boldsymbol{\theta}_1, \boldsymbol{\theta}_2, \ldots, \boldsymbol{\theta}_L)$ denote the weights of each layer $l$. If we denote the parameters in terms of weight and bias matrices of each layer with $\text{in}_l$ input and $\text{out}_l$ output dimensions, $\text{vec}(\boldsymbol{\theta}_l) = [\boldsymbol{W}_l \quad \boldsymbol{b}_l] \in \mathbb{R}^{\text{out}_l \times (\text{in}_l + 1)}$, we can equivalently write this posterior as a factorised matrix normal distribution:

$$q(\boldsymbol{\theta}) = \prod_{i=1}^{N} q(\boldsymbol{\theta}_l), \text{ with} \qquad q(\boldsymbol{\theta}_l) = \mathcal{MN}([\boldsymbol{W}_l \quad \boldsymbol{b}_l] \mid, \boldsymbol{M}_l, \boldsymbol{S}_l \otimes \boldsymbol{A}_l) \qquad (16)$$

The variational parameters $\{\boldsymbol{W}_l, \boldsymbol{S}_l, \boldsymbol{A}_l\}$ provide the mean $\boldsymbol{M}_l$ as well as correlations between layer inputs and bias $\boldsymbol{A}_l \in \mathbb{R}^{(\text{in}_l + 1),(\text{in}_l + 1)}$ and layer outputs $\boldsymbol{S}_l \in \mathbb{R}^{\text{out}_l, \text{out}_l}$. For $L$ hidden layers of width $H$, the number of variational parameters scales quadratically $\mathcal{O}(LH^2)$ compared to the quartic number of variational parameters $\mathcal{O}(LH^4)$ we would need to represent the full covariance. This strikes a practical balance between taking important correlations into account while avoiding having to make a mean-field assumption. Further, we note that the matrix gaussian posterior of [Louizos and Welling, 2016] is the same approximate distribution as used in Kronecker-factored Laplace approximations [Grosse and Martens, 2016]. In Laplace approximations the covariance is the inverse Hessian, whereas in variational inference the covariance is optimised using the ELBO. Layer-factored matrix normal distributions have been succesfully applied to perform approximate Bayesian model selection based on the Laplace approximation in [Immer et al., 2021, 2022, van der Ouderaa et al., 2024]. This work provides evidence that variational inference can also be used to obtain approximate posteriors of this form and obtain a lower bound on the marginal likelihood that is sufficiently tight to perform Bayesian model selection in deep neural networks.

## D.2  Closed-form output variance

It can be shown that the output variance that maximises the marginal likelihood $\hat{\sigma}^2_{\text{data}}$ is the empirical variance of the output. We, therefore, either fixing the output variance $\hat{\sigma}^2_{\text{data}}$ - typically to a very small number in noise-free settings, or setting the output variance to an empirical output variance. An exponentially weighted average of the empirical variance over mini-batches can be used.

**On downscaling the KL term by a $\beta-$scalar**  Many deep learning papers that use variational inference down-scale the KL term by a $\beta-$parameter [Higgins et al., 2017]. We note that, for standard Gaussian likelihoods, scaling the output variance is equivalent to inversely scaling the KL term. We do advice against downscaling of the KL term, as it makes it less clear that the resulting objective is still a lower bound to the marginal likelihood, and hides the fact that the lower bound corresponds to a changed model with altered output variance. In MAP estimation under a Gaussian likelihood, the output variance is arbitrary as it only scales the objective not effecting the optimum, and the objective is often simplified as the mean squared error. In variational inference, the output variance does play an important role of balancing the relative importance between the log likelihood (data fit) and KL term (pull to prior). In this setting, using half mean squared error effectively corresponds to an output variance of $\sigma^2_{\text{data}} = 1$. In practice, this value is often too high because common machine learning datasets have little label noise. As a result, the log likelihood term is too weak and the KL term is too strong. We hypothesise that this has led to practitioners to down-weighting the KL term to obtain sensible posterior predictions, without necessarily realising that they were effectively altering the output variance of the model. Using automatic output variance, the optimal $\hat{\beta}$ can be set to (a running estimate of) the inverse of the empirical variance, also known as the empirical prior precision.

## D.3 Closed-form prior variance minimising inverse KL

Consider the setting of a $D$-dimensional Gaussian $q$, parameterised by mean $\boldsymbol{m}$ and covariance $\boldsymbol{S}$, and a zero mean Gaussian $p_v$ with scalar variance $v$ in each of the equally many dimensions:

$$q = \mathcal{N}(\boldsymbol{m}, \boldsymbol{S}), \qquad\qquad p_v = \mathcal{N}(\boldsymbol{0}, v\boldsymbol{I})$$

where $\boldsymbol{0}$ denotes a zero vector and $\boldsymbol{I}$ an identity matrix. As the log likelihood does not depend on $v$, we can find $v$ that optimises the marginal likelihood by finding the minimiser of the inverse KL:

$$\arg\min_v \mathrm{KL}\left[q \,||\, p_v\right] = \arg\min_v \frac{1}{2}\left[\log\frac{|v\boldsymbol{I}|}{|\boldsymbol{S}|} - D + \mathrm{Tr}((v\boldsymbol{I})^{-1}\boldsymbol{S}) + (\boldsymbol{0}-\boldsymbol{m})^T(v\boldsymbol{I})^{-1}(\boldsymbol{0}-\boldsymbol{m})\right]$$

$$= \arg\min_v \frac{1}{2}\left[\log\frac{|v\boldsymbol{I}|}{|\boldsymbol{S}|} - D + \mathrm{Tr}(\boldsymbol{S})/v + \boldsymbol{m}^T\boldsymbol{m}/v\right]$$

$$= \arg\min_v \left[D\log(v) + \mathrm{Tr}(\boldsymbol{S})/v + \boldsymbol{m}^T\boldsymbol{m}/v\right]$$

Setting the derivative to zero:

$$0 = \frac{\partial}{\partial v}\left[D\log(v) + \frac{1}{v}\mathrm{Tr}(\boldsymbol{S}) + \boldsymbol{m}^T\boldsymbol{m}/v\right]$$

$$0 = -\frac{-Dv + \mathrm{Tr}(\boldsymbol{S}) + \boldsymbol{m}^T\boldsymbol{m}}{v^2}$$

$$v_* = \frac{\mathrm{Tr}(\boldsymbol{S}) + \boldsymbol{m}^T\boldsymbol{m}}{D} \tag{17}$$

We found KL-minimising variance $v_*$ in closed-form as a function of $\boldsymbol{m}$ and $\boldsymbol{S}$. Verified numerically.

## D.4 Plugging minimising variance into KL

Plugging $v_*$ back into the Gaussian $p_{v_*}$ and computing the KL:

$$\mathrm{KL}\left[q \,||\, p_{v_*}\right] = \frac{1}{2}\left[\log\frac{|v_*\boldsymbol{I}|}{|\boldsymbol{S}|} - D + \mathrm{Tr}((v_*\boldsymbol{I})^{-1}\boldsymbol{S}) + (\boldsymbol{0}-\boldsymbol{m})^T(v_*\boldsymbol{I})^{-1}(\boldsymbol{0}-\boldsymbol{m})\right]$$

$$= \frac{1}{2}\left[\log\frac{|v_*\boldsymbol{I}|}{|\boldsymbol{S}|} - D + \frac{D\mathrm{Tr}(\boldsymbol{S})}{\mathrm{Tr}(\boldsymbol{S}) + \boldsymbol{m}^T\boldsymbol{m}} + \frac{D\boldsymbol{m}^T\boldsymbol{m}}{\mathrm{Tr}(\boldsymbol{S}) + \boldsymbol{m}^T\boldsymbol{m}}\right] = \frac{1}{2}\left[\log\frac{|v_*\boldsymbol{I}|}{|\boldsymbol{S}|}\right]$$

$$= \frac{1}{2}\left[D\log\left(\frac{\mathrm{Tr}(\boldsymbol{S}) + \boldsymbol{m}^T\boldsymbol{m}}{D}\right) - \log|\boldsymbol{S}|\right]$$

shows that the resulting KL is only measuring the relative volume $\frac{1}{2}\log\frac{|q|}{|p|}$ between the prior and the posterior, which can be further simplified as

$$\mathrm{KL}\left[q \,||\, p_{v_*}\right] = \frac{1}{2}\left[D\log(\mathrm{Tr}(\boldsymbol{S}) + \boldsymbol{m}^T\boldsymbol{m}) - D\log(D) - \log|\boldsymbol{S}|\right]$$

In practice, we might reparameterise $\boldsymbol{S} = \boldsymbol{L}\boldsymbol{L}^T$ in terms of its triangular Cholesky factor $\boldsymbol{L}$ and use

$$\log|\boldsymbol{S}| = \log|\boldsymbol{L}\boldsymbol{L}^T| = \log|\boldsymbol{L}|^2 = 2\log|\boldsymbol{L}| = 2\log\prod_i \boldsymbol{L}_{ii} = 2\sum_i \log\boldsymbol{L}_{ii}$$

$$\mathrm{Tr}(\boldsymbol{S}) = \mathrm{Tr}(\boldsymbol{L}\boldsymbol{L}^T) = \sum_{i,j}\boldsymbol{L}_{ij}^2$$

This gives the final expression of the KL

$$\mathrm{KL}\left[q \,||\, p_{v_*}\right] = \frac{1}{2}\left[D\log\left(\sum_{i,j}\boldsymbol{L}_{ij}^2 + \sum_i \boldsymbol{m}_i^2\right) - D\log(D) - 2\sum_i \log\boldsymbol{L}_{ii}\right] \tag{18}$$

which implicitly uses the derived optimal variance $v_* = \frac{\sum_{i,j}\boldsymbol{L}_{ij}^2 + \sum_i \boldsymbol{m}_i}{D}$.

# E  Code and Questions

The code is available at `https://github.com/tychovdo/noethers-razor`.
For any questions, please contact the corresponding author, Tycho van der Ouderaa, by email.

