# OpenReview forum: "Noether's Razor: Learning Conserved Quantities"
_NeurIPS.cc/2024/Conference — NeurIPS 2024 poster_

### Official Review · Reviewer_tVqw · 2024-07-08

**Soundness:** 3
**Presentation:** 3
**Contribution:** 1
**Rating:** 5
**Confidence:** 4

**Summary:**

This work proposes a novel way to parametrise Hamiltonians that are approximately invariant to a set of learnable symmetries with the aim of improving existing methods of learning Hamiltonians of dynamical systems. It does so by forcing Hamiltonian neural nets (HNN) to adhere to symmetries, just as in reality, through Noether's theorem. Specifically, they parametrise conserved quantities as quadratic functions for which the flow can be analytically solved and approximately integrate a non-invariant parametrisation of the Hamiltonian (given by an existing HNN) over this flow. The result of this integration is then known to be approximately invariant to the symmetries represented by the quadratic conserved quantities. To optimise their model, which produces an intractable marginal likelihood, the authors resort to variational inference as was done in [1]. In effect, they apply the methodology of [1] to a learnable set of conserved quantities and show experimentally that it allows them to learn the underlying symmetries of different dynamical systems.

[1] van der Ouderaa, Tycho FA, and Mark van der Wilk. "Learning invariant weights in neural networks." Uncertainty in Artificial Intelligence. PMLR, 2022.

**Strengths:**

1. The structure and writing of the paper is fairly good and makes for an easy-to-follow story. I appreciate the completeness of the preliminaries as it makes for a self-contained and clear paper.

2. The goal and main research question that the paper tackles is also very clear from the start: How to learn a representation of the Hamiltonian of a dynamical system that also adheres to symmetry, just like real Hamiltonians. The motivation as to why this is important is also clearly illustrated by the experiments, showing how representations without symmetry (regular HNNs) fail to model good Hamiltonians.

3. The evaluation of how well the learned conserved quantities relate to the true underlying symmetries is reasonably well done. Moreover, it does seem to indicate that the proposed method can learn good representations of symmetries despite the use of approximations.

**Weaknesses:**

1. While the paper is well-structured and self-contained, it is not very clear which parts are background and which parts are the novel contributions. In particular, going through the related work, it seems the only real novelty is the parametrisation of symmetries of H via quadratic conserved quantities (Section 3.1).

2.  The main weakness seems to be the novelty of the paper. Apart from providing a way to parametrise conserved quantities, the rest of the utilised methodology is very close to [1]. To be specific, Section 2 consists of the necessary preliminaries while Section 3.2 explains a well-known procedure to approximately symmetrise a non-invariant function [1]. Section 3.1 is surely novel, but it could be made clearer why quadratic conserved quantities were chosen, i.e. because they lead to analytical flows that are necessary in the symmetrisation. Finally, most of Section 4 and the corresponding appendices are essentially equivalent to the derivations given in Sections 3.6 and 3.7 of [1]. Even the same notation is (wrongly) used in the proof given in Appendix A.1: $E_{\tau} = E_{\tau}\sim E_{\prod_{s = 1}^S p(\tau)}$, which should probably be $E_{\tau} = E_{\prod_{s = 1}^S p(\tau)}$.

3. The proposed metrics to measure how similar the learned symmetries are to the ground truth symmetries make sense, but I have some doubts about the "parallelness" metric. Specifically, the use of the Euclidean norm can give a skewed perspective as it is not an ideal metric for higher-dimensional vectors [2]. Are there no established methods to measure closeness to symmetries? If not, then the proposed metrics are another contribution. If yes, then please cite the necessary literature.

4. The experimental evaluation is lacking variability metrics in general, making it very hard to grasp whether the differences are actually statistically significant and meaningful. Please include either standard error, standard deviation or at least 25-75 percent quantiles over a repeated set of independent runs. Secondly, I wonder why there aren't more baselines for the experiments, especially since the authors mention a couple of competitors [3, 4]. If they are not applicable, then this should be made more explicit as it can also help strengthen the proposed method.

5. I do not agree that the use of approximate Bayesian inference in the context of model selection for deep learning has not been studied/scaled to neural networks as stated in the conclusion (lines 359-360). I would argue that any application of variational inference to Bayesian neural networks (BNN) for example is always a form of model selection as it selects a simpler, yet performant approximation to the BNN within the provided variational family [5, 6].

[1] van der Ouderaa, Tycho FA, and Mark van der Wilk. "Learning invariant weights in neural networks." Uncertainty in Artificial Intelligence. PMLR, 2022.

[2] Aggarwal, C. C., Hinneburg, A., & Keim, D. A. (2001). On the surprising behavior of distance metrics in high dimensional space. In Database theory—ICDT 2001: 8th international conference London, UK, January 4–6, 2001 proceedings 8 (pp. 420-434). Springer Berlin Heidelberg.

[3] Immer, A., van der Ouderaa, T., Rätsch, G., Fortuin, V., & van der Wilk, M. (2022). Invariance learning in deep neural networks with differentiable laplace approximations. Advances in Neural Information Processing Systems, 35, 12449-12463.

[4] Bondesan, R., & Lamacraft, A. (2019). Learning symmetries of classical integrable systems. arXiv preprint arXiv:1906.04645.

[5] Chen, L., Tao, C., Zhang, R., Henao, R., & Duke, L. C. (2018, July). Variational inference and model selection with generalized evidence bounds. In International conference on machine learning (pp. 893-902). PMLR.

[6] Graves, A. (2011). Practical variational inference for neural networks. Advances in neural information processing systems, 24.

**Questions:**

1. Can you more clearly separate novelty from background? Right now, a large part of the paper (Section 4) is discussed as if it were a contribution, while it is more of an application of previous derivations. I would also suggest the authors to perhaps write the paper from a more experimental perspective if it turns out the theoretical contributions are not as extensive.

2. Can you elaborate on the proposed evaluation metrics for symmetry comparisons? Moreover, can you explain why there are no other baselines, such as the Laplace approximations or any of the other methods discussed in Section 2.4? Are they not applicable? If so, accentuate why they are not applicable in the text.

3. Can you provide mean and variability metrics in addition to the given values for Tables 1, 2 and 3? Given the lacking novelty and dependence on empirical results, this is a crucial oversight to me. The experiments should be repeated for multiple seeds.

4. Smaller question out of curiosity: it is mentioned that in Section 5.2 (lines 301-302) that the learned Hamiltonian does not generalise to regions further away from the origin due to a lack of data. The symmetries do seem to improve the generalising behaviour, yet are still insufficient, are there any directions for future work to enforce further physical properties to improve out-of-distribution generalisation? If so, an outlook to such a direction could be interesting.

**Limitations:**

The authors briefly mention that the experiments are rather small scale, but only in the author checklist. It would be appreciated if this is mentioned in the main body of the paper as well. Additionally, the remark on lines 301-302 about generalisation shortcomings could also be highlighted. Finally, the lack of strong guarantees about the quality of the selected/optimised model due to the use of variational inference could also be mentioned, i.e. VI does not guarantee in any way that the variational posterior is in fact invariant or even close to invariance to the learned symmetries. Only experimental evidence can be given in this regard, which luckily shows positive results.

---

> ### Author Rebuttal · Authors · 2024-08-07
>
> Thank you for your feedback and help to improve the paper. We thank the reviewer for finding the paper well-structured and self-contained with a clear goal. The reviewer appreciated the empirical validation and experiments that demonstrate improved generalization of the proposed approach.
>
> > Q1. Novelty
>
> Firstly, most works on symmetry learning focus on relatively simple groups on static domains (e.g. affine invariances on images), whereas we tackle the more novel and open problem of learning symmetries in dynamical systems using train data. The Noether's theorem parameterization of symmetries as conserved quantities in Sec. 3.1 is indeed an important contribution to achieve this. Further, we took considerable steps to actually scale model selection with VI to DNNs, through non-mean-field posteriors and optimized prior/output variances (details in App. D). This differs greatly from typical mean-field and fixed variances commonly used in Bayesian NNs, which have not been shown to capable of performing model selection. Most importantly, we demonstrate successfully learning symmetries of dynamical systems, which are demonstrably correct, through conservation laws in dynamical systems. We deem this a very novel contribution.
>
> > Q2.a. Metrics
>
> We thank the reviewer for pointing out potential nuances in measuring the learned symmetries. First, we’d like to point out that due to the assumption of quadratic conserved quantities, their dimensionality is only roughly quadratic in the dimensionality of the phase space, which is $(2 \times 5)^2=100$ for the 5-harmonic oscillator, and less for the other experiments. Thus the spaces are not very high dimensional.
> There is no standard way of evaluating learned symmetries. Our symmetry correctness measurement reduces to: measure the degree by which two linear subspaces, each generated by non-independent basis vectors, differ. The approach we proposed seemed like a simple way of measuring this, but welcome suggestions for alternatives.
>
> > Q2.b. Baselines
>
> Most recent work on Bayesian model selection that scales to deep learning has relied on the KFAC-Laplace approximation. Laplace is often easier than VI because the posterior covariance does not need to be optimized but can be obtained by estimating the local curvature through gradients. In our particular case, however, Laplace can not be used straightforwardly (we tried this first!). The technical reason for this is that KFAC requires gradients through our forward-pass which in our model already required gradients through the symplectic form. The required 2nd order gradients are not readily supported efficiently in most deep learning framework, not even in JAX/diffrax which is tailored for higher-order derivatives and differentiable solvers. We did take inspiration from Laplace-based approaches in our use of Kronecker-structured posteriors and optimizing variances (see App. D). We thank reviewers by bringing this up and will include this in the main text.
>
> > Q3. Seeds
>
> We will run all experiments with multiple seeds and provide mean and standard error in the final version.
>
> > Q4. Generalization
>
> In principle, generalisation properties are determined by the prior over functions (which includes choices in the neural network architecture). To obtain better performance regions far away from the data we therefore either need data in these regions, or improve the prior over Hamiltonians. This work provides important insights in automatic model selection to DNNs, and therefore provides a pathway to making this process automatic, without the need for retraining and manual cross-validation.
>
> > Limitations
>
> Thank you for your suggestions, we will make sure these limitations are more clearly highlighted in the final version of the paper.
>
> > First successful model selection in DNNs
>
> We follow what is common in Bayesian statistics and refer to the 'model' as the joint distribution (likelihood + prior). Hence, the variational parameters (\mu, \Sigma) are not considered to be part of the model and we therefore do not consider this to be a form of model selection. The hyperparameter \eta parameterizes symmetries/conserved quantities and is part of the model, impacting the effective prior over Hamiltonians H. We show that we can successfully optimize \eta with the ELBO, which demonstrates model selection in DNNs. Although VI can indeed be scaled to DNNs for uncertainty estimates [1], its use for model selection has not been successfully demonstrated yet, as also evidenced by the following citation from  [1]: “Empirically we found optimising the parameters of a prior (by taking derivatives of Eq.1) to not be useful, and yield worse results.”. We took considerable efforts to improve the tightness of the lower bound to make model selection work, using non-mean-field posterior and optimized variances (see App. D for details). So far, we are not aware of other works that have managed to do model selection in DNNs with VI, and therefore consider this to be significant. We hope this clarifies our claim, and will make sure that sufficient context is provided in the final version.
>
>
> [1] Blundell, et al. "Weight uncertainty in neural network." ICML 2025

---

> > ### Comment · Reviewer_tVqw · 2024-08-11
> > **Acknowledgement of author rebuttal**
> >
> > Thank you for your thoughtful answers to my concerns, please consider my follow-up below.
> >
> > **Novelty.**
> >
> > Thank you for further differentiating your work from previous work. I do now better see how using Noether's theorem allows the proposed method to model more complex symmetry groups by modelling their conserved quantities. This does definitely increase the impact and I also agree with the authors that quadratic forms might look simple, though they can lead to the modelling of very non-trivial symmetry groups. I would urge the authors to heavily emphasise this contribution in the main paper, stressing that previous work only modelled simpler affine groups.
> >
> > With respect to the effort of scaling VI for model selection, could you please specify precisely what you contributed here? From appendix D I can see that you started from the work [1] and found an improved closed expression for a special (?) case where the KL is computed between a non-diagonal Gaussian and a zero mean scalar variance Gaussian. This could be an interesting contribution, but it is hard to gauge why this improvement was necessary or why this special case was chosen. What where the bottlenecks that led you to this result and why do you need it? Additionally, since you argue it is crucial to scale VI for model selection, why is it not given more attention in the main text? Especially considering structured VI is an active area of research [2, 3].
> >
> > Lastly, I do maintain that much of the general methodology of the paper is extremely close to previous work, such as [4]. Specifically, Section 3.2 discusses how to approximate the integral in Equation 5 (Equation 1 in [4]) in the same way as [4] did (Sections 3.4 and 3.5 in [4]). Moreover, the contents of Section 4.2 and 4.3 are also similar to Sections 3.6 and 3.7 of, for example, [4]. For the latter there is a reference, but it is not disclosed how different or similar the expressions are. Using the same methodology is completely fine, but the lack of proper references makes the contents look more novel than they are. Please amend this and properly cite the relevant literature.
> >
> > **Metrics.** While not super high-dimensional, 100 dimensions is also not a small number. I would, for example, be curious how using the cosine similarity (distance) influences the results.
> >
> > Considering the novelty of the paper is now more clear to me, I will increase my score to "5: borderline accept". I am willing to increase my score further if the authors can additionally clearly indicate that part of the methodology has been used in previous works. More clarification on the possible contributions to scaling VI might also positively influence my score.
> >
> > [1] Louizos, C. and Welling, M., 2016, June. Structured and efficient variational deep learning with matrix gaussian posteriors. In International conference on machine learning (pp. 1708-1716). PMLR.
> >
> > [2] Hoffman, M.D. and Blei, D.M., 2015. Structured stochastic variational inference. In Artificial Intelligence and Statistics (pp. 361-369).
> >
> > [3] Lindinger, J., Reeb, D., Lippert, C. and Rakitsch, B., 2020. Beyond the mean-field: Structured deep Gaussian processes improve the predictive uncertainties. Advances in Neural Information Processing Systems, 33, pp.8498-8509.
> >
> > [4] van der Ouderaa, T.F. and van der Wilk, M., 2022, August. Learning invariant weights in neural networks. In Uncertainty in Artificial Intelligence (pp. 1992-2001). PMLR.

---

> ### Author Response · Authors · 2024-08-12
> **Thank you for raising the score. Addressing follow-up questions.**
>
> Thank you for raising the score. We are glad that our rebuttal resolved the most pressing concerns, and the impact of Noether's razor is more clear now. To address your follow-up questions,
>
> > VI for model selection
>
> The main topic of the paper is learning symmetries by optimizing learnable conserved quantities by optimizing the marginal likelihood. The main text therefore focuses on the proposed use of Noether's theorem to parameterize symmetries as conserved quantities, and describes how marginal likelihood optimization yields a Noether's razor effect in the prior that enables symmetry learning. We did put quite some effort in scaling VI to model selection, but view our use of VI ultimately as a *means to an end* to approximate the marginal likelihood in deep networks. For this reason, we include details on VI in the appendix, even though some aspects could also be regarded as contributions. Nevertheless, we highlight being able to perform model selection with VI as a significant achievement in the main text, since this has not yet been demonstrated succesfully for deep neural networks before. We expect the details of our VI training scheme to be of independent interest to some VI practitioners, even though we view these as implementation details in the context of this paper, which is symmetry learning through Noether’s razor. Another reason for discussing this in an appendix is that some components are not direct contributions. Nevertheless, combining them together and applying them to perform model selection is. For instance, our use of structured covariances is not novel on its own (matrix normal posteriors are also proposed in [3]), but using them to obtain a tight enough lower bound to perform model selection is novel. Moreover, this particular choice of structured posterior was based on the insight that the matrix normal posteriors of [3] is equivalent to the covariance of the KFAC approximation, which had been shown capable of performing model selection in deep networks using Laplace. We believe that this connection is not widely recognized by the community. Similarly, optimizing prior variance and output variance is not new. However, we note that a large majority of Bayesian deep learning papers relies on downweighting the KL term by a beta factor, which we argue primarily fixes misspecification in the likelihood variance and can better be resolved by optimising this variance empirically (as we do in closed-form). We hope this clarifies why we share details on VI in an appendix, but do highlight the ability to do model selection in deep neural networks with VI as a significant achievement in the main text.
>
> > Closeness to [4]
>
> Our work indeed builds upon [4] and is similar in that both works propose to learn symmetry through a lower bound on the marginal likelihood. We do feel that our work is a big leap forward compared to this prior work in both a) the parameterization and b) the objective function. Firstly, [4] only learns simple affine transformations of pixel coordinates of the input images (like rotations and translations) in supervised learning context. We consider symmetry learning of phase space in the more complex task of modelling dynamical systems, which requires learning the Hamiltonian and predicting trajectories over time. To do so, we use Noether's theorem to parameterize symmetries in terms of conserved quantities, one of our key contributions (see also our overall response). Secondly, [4] has only successfully demonstrated symmetry learning in single hidden layer neural networks with a bound that only integrates out parameters of the last layer. As such, the bound in this paper is not tight enough to go deeper and [4] was not able to successfully scale to deep neural networks. We use a structured and layer-factored posterior and end up with a significantly different lower bound that does scale. Unlike [4], we are able to demonstrate VI-based model selection in deep neural networks.
>
> > Metrics
>
> We're learning a linear subspace of symmetry generators. Via SVD, we find an orthonormal basis of unit vectors of this space. For each of these basis vectors $v_i$, we compute how much overlap there is with the linear subspace of ground-truth symmetries, with orthonormal basis $w_j$. The spaces fully overlap if for each $i$,
> $$
> \lVert v_i^\parallel\rVert^2=\sum_j \langle v_i, w_j \rangle^2 = 1
> $$
> Note that we’re not comparing the overlap between a single vector $v_i$ to and a vector $w_j$, but instead we measure the degree by which $v_i$ lies in the span of all the $w_j$’s. The inner product in this expression is between two vectors of unit norm, so is equivalent to the cosine similarity.
>
> While we are unaware of standard-practice ways of measuring subspace overlap, we deem our proposed metric using singular vectors to be a natural choice. We welcome suggestions for alternatives. We will emphasise the fact that we need to compare linear subspaces, not vectors, better in the paper.

---

### Official Review · Reviewer_Pckk · 2024-07-11

**Soundness:** 3
**Presentation:** 2
**Contribution:** 2
**Rating:** 5
**Confidence:** 3

**Summary:**

This paper proposes Noether's Lazor, a Bayesian framework incorporating learnable symmetries for Hamiltonian neural networks. It parametrizes a hidden symmetry as a flow (single-parameter group) derived from the system's conserved quantity. The flow is then applied to the Hamiltonian as it transforms the system's states while conserving the quantity. The objective function is given as the ELBO that averages over the transformation, forcing the Hamiltonian to be invariant, corresponding to the symmetry. Numerical experiments show that the proposed method finds true symmetries.

**Strengths:**

1. This study addresses the challenging but essential problem of finding symmetries in a Hamiltonian dynamical system.
1. Extracted symmetries in the experiment (Fig 5) are interpretable and interesting.

**Weaknesses:**

1. It is not mentioned enough how the proposed method differs from previous studies. For example, [Alet et al., 2021] might be one of the most relevant works but only mentioned that it uses a validation set. Also, it says that `Similar lower bounds to invariant models that average over a symmetry group have recently appeared in prior work [van der Ouderaa and van der Wilk, 2021, Schwöbel et al., 2022, Nabarro et al., 2022]`, but the differences are not mentioned.
1. The computational complexity is not presented. Since the computation of a Bayesian method is usually heavy, it would be nice to mention how it is compared to the vanilla HNN.
1. The concepts of Noether's theorem and Occam's Razor do not sound tightly connected. As Theorem 1 says, Noether's theorem implies that a Hamiltonian H is G_O invariant, where G_O is the flow of an observable O, iff O is conserved under the system of H. It seems the authors use this notation as a prior of H to impose symmetries (symmetrization of H). However, it takes the integral over the transformation parameter \tau (5). This integral is independent of what we do to obtain the marginal likelihood --- the latter takes an average over the parameter of H. So, I think we can use the symmetrization of H without the Bayesian manner, and I cannot find any special reason why we have to use Noether's theorem and Occam's Razor simultaneously. (If I say something wrong, please correct me.)

**Questions:**

1. I think we can think of two baselines: (a) Bayesian but not symmetry-adapted, i.e. the objective is given by E_\theta[ loss of HNN ] + KL, and (b) symmetry-adapted but not Bayesian, i.e. the objective is Eq.(8) but no KL and no expectation w.r.t. \theta. What would their performance be in the experiments?
1. Why did you use Bayes instead of CV? What are the pros/cons?
1. Can you measure the wall clock time of the vanilla HNN and the proposed method?

Minor issues:
1. C(x) first appears at line 85 without a formal definition.
1. At line 245, H_{\theta, \eta}( \Phi^{\tau}(x) ) would be H_\theta(\Phi^{\tau}_{\eta}(x)) since the symmetrizing parameter \eta is used in the time evolving operator \Phi but not in H.

**Limitations:**

Limitations are not addressed.

---

> ### Author Rebuttal · Authors · 2024-08-07
>
> Thank you for your feedback and help to improve the paper. We thank the reviewer for finding the paper 'intriguing' and 'conceptually very interesting' and that the empirical evaluation shows that the ideas can be promising
>
> > Difference with prior work
>
> We thank the reviewer for pointing this out and will clearly state differences with prior work.
>
> The reviewer is correct that [Alet et al., 2021] is closely related. There are, however, important differences.
> The most important difference is that we explicitly use the learned conserved quantities to infer the symmetries of the phase space. These symmetries then are used to symmetrize the Hamiltonian. In doing so, our work uses the essence of Noether’s theorem. In contrast, their work only learns conserved quantities, without inferring the resulting symmetries. Secondly, [Alet et al, 2021] models images with low dimensional phase spaces, whereas we consider complex dynamical systems with higher dimensional phase spaces. Lastly, while both works appreciate that maximum likelihood training will not allow learning of symmetries, we use approximate Bayesian model selection using VI whereas they rely on a meta-learning objective. Meta-learning can not readily be combined with an ODE solver (as that would require a 2nd order solver) without considerable memory cost, and requires additional hold-out validation data.
>
> The works [van der Ouderaa and van der Wilk, 2021, Schwöbel et al., 2022, Nabarro et al., 2022] use a similar way of lower bounding learnable symmetries via the ELBO, but only consider simple affine invariances in image classification tasks, whereas we learn more complex symmetry groups in dynamical systems. A second difference is that we use Noether’s theorem to parameterize learnable symmetries in terms of their associated conserved quantities. Lastly, prior works that use VI for symmetry learning have so far only demonstrated this successfully using single-layer approaches (e.g. single layer / last-layer VI), whereas we manage to scale our work to deep neural networks (details in App. D.)
>
> > Connection between Noether's theorem and Occam's razor
>
> While it may not be necessary to learn Hamiltonians symmetrized by conserved quantities via Occam’s razor, we argue that it is very compelling to do so. Learning symmetries naively is plagued by learning trivial symmetries, or only some of the symmetries repeatedly. Occam’s razor via Bayesian model selection provides a single ELBO objective that is simple to optimize. For example, we have shown that our method is able to discover a 25-dimensional symmetry group in the 5-harmonic oscillator. Also, we found all 7 conserved quantities of a 2-body system, generating a much larger group than the 3-dimensional group SE(2) we (naively) expected to find. This shows our method is very successful in avoiding these trivial solutions and instead finds all the symmetries in a system.
>
> > Q1. Baselines
>
> To be clear, we train both the HNN baseline and HNN + symmetry adaptation using the same variational inference scheme (so are both `Bayesian' in that sense) to allow fair comparison. As such, the improvements in test performance can therefore be attributed to the proposed symmetry learning through Noether's razor. We will make sure this is very clear from the main text. All works on symmetry learning that we know of acknowledge the fact that learning symmetries can not reliably be done through maximum likelihood only, and therefore rely on some form of validation data, regularisation, or marginal likelihood estimates to perform model selection. Regardless, we will also run a ('non-Bayesian') maximum likelihood baselines of both HNN and HNN + symmetry adaptation in the final version.
>
> > Q2. Bayes instead of cross-validation
>
> This is a great question. The main benefits of Bayes compared to CV is that the objective is 1) differentiable and we can thus optimize hyperparameters with gradients and 2) we have a single training procedure that does not require any retraining. With CV, we can evaluate only a finite number of settings for the conserved quantities that would have to be specified in advance. Further, it requires retraining the entire model for each possible configuration to be evaluated on hold-out validation data. We thank the reviewer for raising this, as it is an important motivation behind our method which we will make more clear from the text.
>
> > Q3. Wall clock time
>
> We will report the memory and computational cost of our method and baseline in the final version.

---

> > ### Author Response · Authors · 2024-08-12
> >
> > Dear reviewer Pckk,
> >
> > We would appreciate a response to our rebuttal so that we can still address follow-up questions or concerns you might have.
> >
> > Sincerely,
> > Authors

---

> > > ### Comment · Reviewer_Pckk · 2024-08-14
> > > **To authors**
> > >
> > > I appreciate that the authors carefully responded to my review. My concerns regarding prior studies, Q1, and Q2 are now resolved. I hope the authors will address the issue of computational cost. Although I'm not fully convinced about the design choice to combine Noether's theorem and Occam's razor, I admit it works in some scenarios. I will raise my score.

---

### Official Review · Reviewer_zC9J · 2024-07-11

**Soundness:** 3
**Presentation:** 2
**Contribution:** 3
**Rating:** 5
**Confidence:** 4

**Summary:**

This paper proposes a Bayesian framework to learn conserved quantities (and, implicitly, symmetries) in the context of Hamiltonian systems. The idea is that the Hamiltonian system has specific parametric conserved quantity (given by a quadratic function, line 136), in that case the conserved quantity can be learned from data in a Bayesian setting (using the parameterization of the conserved quantity as a prior over the possible Hamiltonian systems) via ELBO.

Even though the setting is a bit restricted, the idea is quite intriguing and conceptually very interesting. My main criticism of the paper is that the methodology is not explained in a clear, self-contained way, making the paper hard to follow for the NeurIPS audience. Since the main contribution of the paper is conceptual, I think this is a major drawback. The paper seems hastily written (it has many typos, including in the abstract "conversation laws" instead of "conservation laws", inconsistent notations, and incomplete definitions) and it could significantly benefit from a careful major revision.

**Strengths:**

The topic is very interesting. Incorporating symmetries via conservation laws in machine learning models is an interesting idea in the context of physics informed machine learning, and machine learning models for physical systems. The combination of Noether's theorem with Bayesian modeling is novel. The empirical evaluation, though limited, show that these ideas can be promising.

**Weaknesses:**

In my opinion, the paper would benefit from a major revision focused on improving the exposition to clearly explain the methodology.

The paper considers a specific example, where the conservation law is parameterized by a quadratic function. Under this assumption we have an explicit formula for the flow $\Phi$ that seems to be key for implementing the optimization. Without this assumption it is unclear to me that the problem is computationally tractable. I don't see this as a weakness necessarily (though it would make sense to state it as a limitation in the conclusion, if that's the case). However, it does make the main contribution of the paper to be mostly conceptual. Therefore this paper main contribution should be the explanation of the methodology. In my opinion this explanation is not sufficiently clear.

In particular it'd be nice to see an explanation of why one would choose to optimize the ELBO over direct optimization over the parameters that define the conservation laws and the Hamiltonian.


**Comments:**
- Line 85: The definition of the bracket depends on C which has not been defined before. Should it be O?
- Please make explicit what is the domain and range of the functions considered in the manuscript. For example
$x: \mathbb R \to \mathcal M$
- This sentence could be explained better: "We have used a different symbol to not conflate the ODE time $\tau$ with regular time $t$ of the trajectory generated by the Hamiltonian."? In particular in relation to referring $\tau$ as "symmetry time".
- Also, the difference on the parameters $\eta$ that define the space of symmetries vs $\theta$ that define the space of Hamiltonians should be made more explicit (section 3.1). In particular, this should be explained in section 4.
- Section "Why the marginal likelihood can learn symmetry". What does it mean to "learn symmetry" mathematically in this context? Does it mean to learn $\eta$?
- A few typos: "conversation laws", "train data", "generalisaiton", "als".

**Questions:**

Is an explicit formula for $\Phi$ needed in order to implement the optimization? Is that a limitation for the approach?

What is the relevance of the distribution over $\tau$? How do the results depend on it?

Can we used this to learn the specific conservation law ($\eta$) or is it only known implicitly in the solution $H(\Phi)$?

**Limitations:**

The methodological limitations are not explicitly stated.

---

> ### Author Rebuttal · Authors · 2024-08-07
>
> Thank you for your feedback and help to improve the paper. We thank the reviewer for finding the paper 'intriguing' and 'conceptually very interesting' and that the empirical evaluation shows that the ideas can be promising
>
> > Q1. Non-quadratic conserved quantity
>
> First, we believe that the quadratic conserved quantity is not an overly strong assumption. It restricts us to the symmetry group having an affine (linear+translation) action on the phase space, but it does not limit the structure of the symmetry group itself. As far as we know, basically any work on in/equivariant deep learning uses groups with an affine action. Please see our overall rebuttal for more details.
>
> If one wishes, our method can be generalized to non-quadratic conserved quantities whose flows have no explicit solution! To optimise our objective we require differentiable samples from our flow. In case of a non-quadratic conserved quantity, we can use the reparameterization-trick by sampling a particular symmetry time and solving the flow through a differentiable ODE solver. Our code, in fact, also includes an implementation of this in JAX (using the diffrax library), which we will release upon acceptance. We chose not to describe this in-depth in the paper, because we relied on quadratic conserved quantities for our main results.
>
> > Q2. Direct optimization of conserved quantities
>
> As described in Sec. 4, we can not directly learn conserved quantities with data fit loss because no symmetry will always make fitting easier (even though it can hurt generalization). This is acknowledged by virtually every symmetry learning paper, which is why most methods rely on validation data or other forms of model selection. Unlike cross-validation, which requires validation data and can only evaluate certain settings of hyperparameters through expensive model retraining, approximate Bayesian model selection allows for gradient-based optimization of hyperparameters.
>
> > Q3. Relevance of distribution over \tau
>
> We only considered a zero-centred Gaussian distribution over \tau with learning the variance and found this to work well across experiments. Although we are free to choose any distributional family, we expect that unimodal distributions around the origin are often the most sensible and least difficult to optimize. Further, we can argue that a Gaussian subsumes a regular HNN, which would be equivalent to having a \delta peak (zero variance Gaussian) at the origin. Taken together, we think a Gaussian \tau is sufficient in most cases, especially since the variance is learned, and will add some of these considerations in the main text.
>
> > Q4. Can we use this to learn the specific conservation law?
>
> Yes! We can directly inspect the learned conservation laws as well as the associated symmetries, which was not possible in prior work. We find that we learn the correct conservation laws for our n-harmonic oscillator and n-body system experiments. With the quadratic conserved quantity, we can compute the symmetry action with a matrix exponential. With a free-form conserved quantity, this would require solving an ODE. This is a clear advantage of our method compared to prior work, and we will make this more clear from the main text.
>
> > Description of our methodology
>
> The reviewer points out that the current methodology is not always clear from the main text. We agree that we might have focused too much on the conceptual idea of Noether's razor and will move more practical details and contributions from the appendices to the main text (e.g. App. D). This should give a more explicit description of how the forward-pass and objective calculation are implemented in practice.
>
> > Comments
>
> We agree that the difference between parameters \theta and symmetry parameters \eta can be made more explicit early on. In our work, we have a neural network for F and a quadratic conserved quantity for C, so \theta are all neural network parameters and \eta are the matrices A and vectors b. With learning symmetries, we mean learning \eta, which parameterise learnable conserved quantities and thus directly imply learnable symmetries by Noether's theorem. We hope this resolves the issue.
>
> Typos will be fixed, thanks for spotting them.

---

> > ### Comment · Reviewer_zC9J · 2024-08-12
> >
> > Thank you for the answers. I suggest you include a discussion on the generality of the quadratic conservation law. I increased my score.

---

### Official Review · Reviewer_qjc1 · 2024-07-12

**Soundness:** 3
**Presentation:** 3
**Contribution:** 3
**Rating:** 6
**Confidence:** 3

**Summary:**

The authors propose to use the parameterized symmetries for learning correct Hamiltonian dynamics from data. It is based on the Noether’s theorem, which states that the continuous symmetries generated by observables O for Hamiltonian H yields the conservation of O, and vice versa. To do this, the authors parameterizes the observables as a quadratic form, and makes the learnable Hamiltonian function (neural network) to be invariance  with respect to the one-parameter group (flow) generated by O, which is analytically expressible. Invariance is achieved through techniques such as orbit pooling, which averages the outputs of the Hamiltonian over the group orbit. To learn the Hamiltonian with proper symmetries, the authors propose to use the Bayesian approach, i.e., learning the ELBO, thus the title of the paper is the “Noether’s razor” (similar to the Occam’s razor that regularize the model with proper weights). The authors validate their approach to classical examples like Harmonic oscillator and n-body systems, which showcase the method can model the appropriate dynamics as well as find a correct symmetries.

**Strengths:**

The motivation of this paper is clear, and highly relevant to both the AI + Science and geometric machine learning communities. The paper is generally well-written and easy to follow. The method is based on well-known Noether’s theorem, thus is very principle. Overall, I believe this paper is worthy of publication and will provide valuable insights to researchers in the relevant field, even though it may not be particularly striking to a broader ML audience.

**Weaknesses:**

- This paper assumes that the symmetry arises from the symplectic flow of a quadratic form, which is a significant assumption that lacks a clear explanation. The authors should justify the use of a quadratic form, demonstrating that it is a reasonable assumption for the systems of greatest interest (not only for the benchmarks used in the paper).

- The experiment consists only of low-dimensional examples modeling phase space trajectories directly, raising doubts about whether the proposed results can be applied to high-dimensional data. Please note that even the original Hamiltonian neural network [1] includes a task involving learning the underlying dynamics from images.

**Questions:**

The authors directly parameterize the model Hamiltonian to be invariant by averaging over the group orbit. Another way to achieve the symmetry is satisfying {C, H} = 0, according to the Noether’s theorem. Since the latter can be achieved easily by adding the regularizer like ||{C, H}||, I am curious about why the authors chose to use the method proposed in the paper instead of this regularizer, and how much of a performance difference there is between the two methods.

Some typos:

- page 2, line 70, $x_t + J \nabla H_{\theta} (x_t) \Delta t$

- page 3, line 85, $\nabla O^T(x) \cdot J \nabla H (x) $

- page 3, line 120, also

- page 4, line 156, the the

**Limitations:**

The authors do not explicitly mention the limitation in the main manuscript. Please refer and address Weaknesses.

---

> ### Author Rebuttal · Authors · 2024-08-07
>
> Thank you for your feedback and help to improve the paper. We thank the reviewer for finding the paper clear, well-written, easy to follow, and highly relevant to both the AI + Science and geometric machine learning communities. Further, we appreciate the reviewers deemed the paper very principled and 'worthy of publication'.
>
> > Quadratic form
>
> We indeed consider conserved quantities of the quadratic form, but feel this is not an overly strong assumption. It restricts us to the symmetry group having an affine (linear+translation) action on the phase space, but it does not limit the structure of the symmetry group itself. As far as we know, basically any work on in/equivariant deep learning uses groups with an affine action. The model can in principle still learn any dynamical system, also those with non-quadratic symmetries, given a sufficiently flexible network. Compared to regular HNN models, which do not have any learnable symmetries built-in, we deem our model as an improvement, even in those cases, as it can improve generalisation by picking up on some quadratic conserved quantities which may be present. Please see our overall rebuttal for more details.
>
> > High-dimensional data
>
> In our experiments, we focus on n-harmonic oscillators in higher-dimensions and chaotic n-body systems, both directly on phase space. We did find that our method remains to work well when increasing dimensionality. We discover a 25-dimensional symmetry group for n=5, which for groups is very high-dimensional.
>
> Our focus is demonstrating the principle of learning symmetrisation through conserved quantities. We therefore do not also consider learning models from images or only observing positional data. These tasks would imply a latent variable model which would significantly complexity the model description and distract from our primary research interest of demonstrating the Noether's razor effect. Regardless, we see no a-priori reason why Noether’s razor could not be applied to such extensions since they have been shown to work for regular HNNs in prior work. We consider this an interesting follow-up research direction, but out of scope for this paper.
>
> > Regularizing objective
>
> This is an interesting suggestion. Instead of averaging the neural network along the group orbit, we indeed could also have considered regularising it in the same direction. In principle, this would also be a valid way of incorporating symmetry in the prior and is common for some other hyperparameters, such as the prior variance on weight magnitudes. We hypothesise that averaging generalises better as it smoothes the function globally, not only around train data, but refrain from making hard statements on this, since we did not try this yet. Although regularization could offer an interesting alternative to averaging to perform symmetrization, our use of averaging should not impact the main conclusions of our work, which was demonstrating Noether's razor. Nevertheless, we agree with the reviewer that it would be interesting to try this and intend to add it as an alternative baseline in the final version.
>
> Typos will be fixed, thanks for spotting them.

---

> > ### Comment · Reviewer_qjc1 · 2024-08-12
> >
> > Thank you for your response. While the authors have addressed some of my concerns, their response is not sufficiently convincing to warrant an increase in the score. I will maintain my positive score of 6.

---

### Official Review · Reviewer_hvbw · 2024-07-22

**Soundness:** 4
**Presentation:** 4
**Contribution:** 4
**Rating:** 9
**Confidence:** 4

**Summary:**

This work aims to model symmetries for strong inductive biases in machine learning models of dynamic systems. Instead of constraining models to certain symmetries, this work focuses on automatically learning them from data. It proposes to parameterize symmetries using conserved quantities by Noether's theorem. The conserved quantities are further incorporated into priors and the model is trained by optimizing a lower bound of the marginal likelihood, which is able to balance both data fit and model complexity. Empirical results on both n-harmonic oscillators and n-body system problems are presented.

**Strengths:**

- The integration of Noether’s theorem with variational inference to learn conserved quantities for symmetries is a novel and significant contribution, providing a new perspective on symmetry learning.
- Experiments show that the learned symmetries have the same rank as the ground truth symmetries, showing the effectiveness of the proposed learning method. Further from the empirical results, the model with learned symmetries has a similar performance to the one with built-in oracle symmetries and outperforms the ones without.
- The work is overall well-written and has provided sufficient background for the authors to understand the proposed method as well as references to existing work.

**Weaknesses:**

- This work can be further improved by providing computational analysis on the computation overhead induced by symmetry learning. Maybe the authors can elaborate on this point.
- From Section 5.1, it seems that the choice of hyperparameter K has to be greater than the dimension of ground-truth symmetry in order to capture the full symmetries. I wonder how to choose such hyperparameters in practice.

**Questions:**

- See weakness.

**Limitations:**

Yes.

---

> ### Author Rebuttal · Authors · 2024-08-07
>
> Thank you for your feedback and help to improve the paper. We appreciate that the reviewer rated our paper with a "9: Very Strong Accept", found our approach a novel and significant contribution, well presented and recognized the new perspective on symmetry learning as well as convincing experimental results.
>
> We agree with the authors that including computational analysis can further strengthen the paper and will report time measurements in the final version. In terms of the choice of hyperparameter K, the singular values described in Sec. 5.2 can be used to validate a chosen setting of K. If there are many (close to) zero singular values, K can be decreased, and if there are none such values, K should be increased. Setting K too high will not hurt performance, but add some computational overhead. In our experiments, we typically set K=10 or K=20.

---

### Author Rebuttal · Authors · 2024-08-07

We thank all reviewers for the feedback and help to improve the paper. We are excited to have received a "9: Very Strong Accept", and are confident that the concerns raised by the lower score reviews are sufficiently addressed in this rebuttal. Overall, most reviewers found the paper very 'clearly written', 'principled', 'easy-to-follow' and 'highly relevant to both the AI + Science and geometric machine learning communities'. In particular, we are proud that reviewers found the paper 'intriguing and conceptually interesting’, and were positive about the empirical validation and deemed the results promising.

Most raised concerns are addressed in direct rebuttals. Since several reviewers raised questions about the use of the quadratic form, we also provide a more thorough discussion on this issue in this overall rebuttal:

> Quadratic form

We indeed consider learnable conserved quantities of the quadratic form. Regardless of the quadratic form, our model can in principle still learn any dynamical system, also those with non-quadratic symmetries, given a sufficiently flexible network. This is already a direct improvement over models without learnable invariance properties, such as regular HNNs, while also capturing commonly encountered symmetries that are currently hard-wired into architectures.

Regarding the quadratic form as a type of learnable symmetry, we feel that this is not an overly strong assumption. Quadratic conserved quantities can represent any symmetry that has an affine (linear+translation) action on the state space. This includes essentially all cases that are currently studied in geometric deep learning. Note that the quadratic constraint does not limit the shapes of the symmetry groups that we learn. As an example, in our experiments we find the Euclidean group, which is itself a curved manifold with non-trivial topology. Hence, we find this quadratic assumption to be not overly strong. However, we welcome suggestions for common examples in deep learning with equi/invariance to groups with a non-affine action, so that we can list these examples as limitations of our method.

The technical reason why quadratic conserved quantities can represent complex groups, is that the learned conserved quantities are a basis of the Lie algebra of the Lie group of symmetries, with the Poisson bracket being identical to the Lie bracket of the Lie algebra. The structure of the Poisson bracket between the conserved quantities determines the shape of the group, and this is not constrained by the conserved quantities being quadratic.

If one wishes, we can extend our framework to non-quadratic conserved quantities to model non-affine actions. This has the downside that the symmetrization needs to be done with an ODE-solving, as in Eq. (3), instead of with the matrix exponential. This slows down the implementation by a large amount. Furthermore, such free-form conserved quantities have many more parameters, which complicates the Bayesian model selection. We did implement this and will include it in the code that will be released upon acceptance. We have omitted the details on free-form conserved quantities from the present manuscript for space reasons, but will elaborate on it in the appendix of the final version.

---

### Author Response · Authors · 2024-08-07
**Overall rebuttal**

We thank all reviewers for the feedback and help to improve the paper. We are excited to have received a "9: Very Strong Accept", and are confident that the concerns raised by the lower score reviews are sufficiently addressed in this rebuttal. Overall, most reviewers found the paper very 'clearly written', 'principled', 'easy-to-follow' and 'highly relevant to both the AI + Science and geometric machine learning communities'. In particular, we are proud that reviewers found the paper 'intriguing and conceptually interesting’, and were positive about the empirical validation and deemed the results promising.

Most raised concerns are addressed in direct rebuttals. Since several reviewers raised questions about the use of the quadratic form, we also provide a more thorough discussion on this issue in this overall rebuttal:


> Quadratic form
We indeed consider learnable conserved quantities of the quadratic form. Regardless of the quadratic form, our model can in principle still learn any dynamical system, also those with non-quadratic symmetries, given a sufficiently flexible network. This is already a direct improvement over models without learnable invariance properties, such as regular HNNs, while also capturing commonly encountered symmetries that are currently hard-wired into architectures.

Regarding the quadratic form as a type of learnable symmetry, we feel that this is not an overly strong assumption. Quadratic conserved quantities can represent any symmetry that has an affine (linear+translation) action on the state space. This includes essentially all cases that are currently studied in geometric deep learning. Note that the quadratic constraint does not limit the shapes of the symmetry groups that we learn. As an example, in our experiments we find the Euclidean group, which is itself a curved manifold with non-trivial topology. Hence, we find this quadratic assumption to be not overly strong. However, we welcome suggestions for common examples in deep learning with equi/invariance to groups with a non-affine action, so that we can list these examples as limitations of our method.

The technical reason why quadratic conserved quantities can represent complex groups, is that the learned conserved quantities are a basis of the Lie algebra of the Lie group of symmetries, with the Poisson bracket being identical to the Lie bracket of the Lie algebra. The structure of the Poisson bracket between the conserved quantities determines the shape of the group, and this is not constrained by the conserved quantities being quadratic.

If one wishes, we can extend our framework to non-quadratic conserved quantities to model non-affine actions. This has the downside that the symmetrization needs to be done with an ODE-solving, as in Eq. (3), instead of with the matrix exponential. This slows down the implementation by a large amount. Furthermore, such free-form conserved quantities have many more parameters, which complicates the Bayesian model selection. We did implement this and will include it in the code that will be released upon acceptance. We have omitted the details on free-form conserved quantities from the present manuscript for space reasons, but will elaborate on it in the appendix of the final version.

---

### Decision · Program_Chairs · 2024-09-25

**Decision:**

Accept (poster)

**Comment:**

This paper presents an innovative approach that integrates Noether’s theorem with Bayesian model selection to learn conserved quantities directly from data. This method enables the automatic discovery of symmetries in dynamical systems, which is both novel and impactful. I recommend accepting the paper for the following reasons. The primary strength of this work lies in its successful combination of Noether’s theorem with variational inference to learn conserved quantities, which, in turn, defines the symmetries of the system. This approach addresses a significant challenge in machine learning and physics-informed models, where incorporating symmetries can greatly enhance model performance and generalization. The paper's method is particularly powerful because it avoids the need for manually designed symmetries or additional regularizers, instead relying on the natural outcome of the Bayesian model selection process. The authors demonstrate the effectiveness of their approach through empirical results on classical systems like harmonic oscillators and n-body problems, where the learned symmetries match the ground truth, further validating the method. The paper is also well-written, clearly explaining the methodology and its implications for both AI and physics communities. The authors have effectively addressed reviewer concerns, particularly those related to the choice of quadratic forms and the scalability of their approach to more complex, high-dimensional systems. Their response to these concerns, including additional details in the appendix, strengthens the overall contribution. While some concerns were raised about the computational complexity and the specific assumptions made (e.g., quadratic forms for conserved quantities), these do not significantly detract from the overall quality of the work. The authors have provided sufficient justification and context, making a strong case for the utility and novelty of their approach. Given its strong theoretical foundation, practical relevance, and positive empirical results, this paper is a valuable contribution to the field and should be accepted.